# The selenocysteine-containing protein SELENOT maintains dopamine signaling in the midbrain to protect mice from hyperactivity disorder

Qing Guo[1,2], Zhao-Feng Li[1], Dong-Yan Hu[1], Pei-Jun Li[2], Kai-Nian Wu[1], Hui-Hui Fan[1], Jie Deng[3], Hong-Mei Wu[1], Xiong Zhang [ID][2✉] & Jian-Hong Zhu [ID][1,2✉]

## Abstract

**Dopaminergic neuron dysfunction has been implicated in multiple neurological and psychiatric disorders. SELENOT is a selenocysteine-containing protein of the ER membrane with anti-oxidant and neuroprotective activities, but its pathophysiological role in dopaminergic neurons remains unclear. In this study we show that male mice with SELENOT-deficient dopaminergic neurons exhibit attention deficit/hyperactivity disorder (ADHD)-like symptoms, including hyperlocomotion, recognition memory deficits, repetitive movements, and impulsivity. Dopamine metabolism, extrasynaptic dopamine levels, spontaneous excitatory postsynaptic currents in the striatum, and electroencephalography theta power are all enhanced in these animals, while dopaminergic neurons in the substantia nigra are slightly reduced but with normal firing and cellular stress levels. Our results also indicate that the expression of dopamine transporter (DAT) is significantly reduced in the absence of SELENOT. Both the development of ADHD-like phenotypes and DAT downregulation are also observed when SELENOT is absent from the whole brain, but not when its conditional knockout is restricted to astrocytes. Mechanistically, we show that SELENOT downregulates DAT expression via interaction with SERCA2 of the ER -but not with IP3R or RYR- to regulate the ER-cytosol $Ca^{2+}$ flux and, subsequently, the activity of transcription factor NURR1 and the expression levels of DAT. Treatment with amphetamine or methylphenidate, which are commonly used to treat ADHD, reverses the hyperactivity observed in mice with SELENOT-deficient dopaminergic neurons. Our study demonstrates that SELENOT in mouse dopaminergic neurons maintains proper dopamine signaling in the midbrain against the development of ADHD-like behaviors.**

**Keywords** SELENOT; Dopaminergic Neurons; Attention Deficit/ Hyperactivity Disorder; Dopamine Transporter; Calcium
**Subject Categories** Metabolism; Neuroscience; Signal Transduction

## Introduction

Midbrain dopaminergic neurons are abundantly located in the substantia nigra pars compacta (SNpc) and the ventral tegmental area (VTA). The nigrostriatal projection of dopaminergic neurons from the SNpc to the dorsal striatum mainly controls voluntary movement as part of the basal ganglia circuit. The mesolimbic projection of dopaminergic neurons from the VTA to the ventral striatum and the mesocortical projection from the VTA to the prefrontal cortex regulate multiple functions, including reward, addiction, emotion, and motivation. Striatum-derived postsynaptic dopamine mediates nigrostriatal and corticostriatal transmission in medium spiny neurons bearing dopamine D1 receptor (D1R) and/or dopamine D2 receptor (D2R), thereby controlling movement and some cognitive functions (Haber et al, 2000; Luo and Huang, 2016). Defects in midbrain dopaminergic neurons have been implicated in multiple neurological and psychiatric disorders, such as Parkinson's disease, schizophrenia, attention deficit/hyperactivity disorder (ADHD), and drug addiction (Conio et al, 2020; Klein et al, 2019).

Selenoprotein T (SELENOT), a selenocysteine (Sec)-containing and evolutionarily conserved protein, is anchored to the endoplasmic reticulum (ER) membrane through a hydrophobic domain (Gladyshev et al, 2016; Grumolato et al, 2008; Zhang et al, 2019). SELENOT possesses thioredoxin reductase-like activity centered on the Sec residue (Boukhzar et al, 2016; Labunskyy et al, 2014) and is implicated in cytosolic $Ca^{2+}$ maintenance in PC12 cells and SK-N-SH cells, potentially via Sec-mediated redox regulation (Grumolato et al, 2008; Shao et al, 2019). Whole-body *Selenot* deficiency leads to early embryonic lethality in mice (Boukhzar et al, 2016). The mouse Encyclopedia of DNA Elements Project predicts that SELENOT is preferentially expressed in the central nervous system, cerebellum, and kidney (Wang et al, 2019). SELENOT is known to regulate hormone maturation and secretion in endocrine cells (Hamieh et al, 2017) and to safeguard dopaminergic neurons in cellular and mouse models of Parkinson's disease (Boukhzar et al, 2016; Shao et al, 2019). However, the pathophysiological role of SELENOT in dopaminergic neurons remains elusive.

ADHD is a prevalent neurodevelopmental disorder characterized by predominant symptoms in the domains of inattention,

[1]Institute of Nutrition and Diseases and Center for Research, School of Public Health, Wenzhou Medical University, Wenzhou, China. [2]Department of Neurology and Institute of Geriatric Neurology, the Second Affiliated Hospital and Yuying Children's Hospital, Wenzhou Medical University, Wenzhou, China. [3]Zhejiang Key Laboratory of Intelligent Cancer Biomarker Discovery and Translation, The First Affiliated Hospital, Wenzhou Medical University, Wenzhou, China. ✉E-mail: xiongzhang@wmu.edu.cn; jhzhu@wmu.edu.cn

hyperactivity, and impulsivity, either individually or combined, and primarily occurs in children and often extends into adolescence and adulthood (Gallo and Posner, 2016; Posner et al, 2020). While the etiology of ADHD is not fully understood, its pathophysiology has been linked to frontoparietal, dorsal frontostriatal, and mesocorticolimbic circuits (Gallo and Posner, 2016). In this study, we demonstrated the essential roles of dopaminergic neuron SELENOT in preventing ADHD-like behaviors. The underlying mechanisms involved the SELENOT-mediated nigral ER-cytosol $Ca^{2+}$ homeostasis, dopamine transporter (DAT), and striatal dopamine transmission.

## Results

### *Selenot*<sup>fl/fl</sup>;*Dat-cre* mice display ADHD-like behaviors including hyperlocomotion, recognition memory deficits, repetitive movements, and impulsivity

Conditional knockout of *Selenot* in dopaminergic neurons (*Selenot*<sup>fl/fl</sup>;*Dat-cre*) was generated through *Dat* promoter-mediated CRE excision (Fig. 1A) and confirmed by co-immunostaining of SELENOT and tyrosine hydroxylase (TH) in the substantia nigra (Fig. 1B,C). SELENOT was present nearly in all dopaminergic neurons and showed no expression difference between the substantia nigra and VTA (Fig. 1B–D). The development of *Selenot*<sup>fl/fl</sup>;*Dat-cre* mice and *Selenot*<sup>fl/fl</sup> control mice was comparable, as indicated by 1) the cortex and hippocampus volumes at postnatal day 3 and week 6 (Appendix Fig. S1), and 2) body weight at week 8 (Fig. 1E). Results from the open field test (Fig. 1F) showed that spontaneous locomotion (distanced traveled every 10 min and total distance; Fig. 1G,H), active time, and mean moving speed (Fig. 1I,J) were significantly elevated in *Selenot*<sup>fl/fl</sup>;*Dat-cre* mice compared to *Selenot*<sup>fl/fl</sup> mice. These two lines of mice exhibited comparable amounts of center distance and center time in the open field (Appendix Fig. S2A,B), suggesting no anxiety in *Selenot*<sup>fl/fl</sup>;*Dat-cre* mice. Results from the elevated plus maze test confirmed hyperlocomotion (increased total entries to arms; Fig. 1K) and no anxiety (increased entries to open arms but comparable time spent in open arms; Fig. 1L,M) in *Selenot*<sup>fl/fl</sup>;*Dat-cre* mice.

Compared to *Selenot*<sup>fl/fl</sup> mice, *Selenot*<sup>fl/fl</sup>;*Dat-cre* mice showed: (1) impairment in short-term (1 h; Fig. 1O,P; Appendix Fig. S2C) and long-term memory (24 h; Fig. 1Q,R; Appendix Fig. S2D) as assessed by the novel object recognition test (Fig. 1N), (2) repetitive behaviors in rearing, tail flicking, and climbing (Fig. 1S), and (3) enhanced impulsive behavior based on the cliff avoidance reaction test (Fig. 1T). In contrast, *Selenot*<sup>fl/fl</sup>;*Dat-cre* mice appeared normal in motor coordination, measured according to the rotarod test (Appendix Fig. S2E,F), spatial memory, evaluated using the Y-maze test (Appendix Fig. S2G), frequencies of self-grooming, jumping, and digging (Fig. 1S), and social interactions, assessed by the three-chamber sociability test, including social preference (Appendix Fig. S2H–K) and social recognition (Appendix Fig. S2L–O).

### Amphetamine or methylphenidate treatment, or SELENOT overexpression, rescues the hyperactive phenotype in *Selenot*<sup>fl/fl</sup>;*Dat-cre* mice

Two psychostimulants commonly used to treat ADHD, amphetamine and methylphenidate (Cortese et al, 2018), were used to

investigate their efficacy in alleviating *Selenot* deficiency-induced ADHD-like symptoms. Locomotion in the *Selenot*<sup>fl/fl</sup> control mice was not sensitized by treatment with amphetamine (1 or 2 mg/kg), but it was sensitized by methylphenidate (15 or 30 mg/kg; Fig. 1U–X; Appendix Fig. S2P–S). These results were similar to the reported effects of the psychostimulants on wild-type mice with these dosages (Mines et al, 2013). Treatment with amphetamine at 2 mg/kg (Fig. 1U,V) or methylphenidate at 30 mg/kg (Fig. 1W,X), but not at lower dosages (Appendix Fig. S2P–S), rescued the hyperlocomotion phenotype in *Selenot*<sup>fl/fl</sup>;*Dat-cre* mice. These findings, with pharmacological implications, further corroborate the novel role of SELENOT in protecting against ADHD-like behaviors.

Furthermore, to determine whether recovering SELENOT expression rescues the hyperactivity observed in *Selenot*<sup>fl/fl</sup>;*Dat-cre* mice, we in vivo overexpressed SELENOT in dopaminergic neurons of the substantia nigra using a TH promoter-initiated viral vector (Fig. 1Y,Z). Similar to the effect of amphetamine, SELENOT overexpression did not appear to affect locomotion in *Selenot*<sup>fl/fl</sup> control mice. In contrast, the overexpression successfully rescued the hyperlocomotion phenotype in *Selenot*<sup>fl/fl</sup>;*Dat-cre* mice (Fig. 1AA,AB).

### *Selenot* deficiency in the whole brain, but not in astrocytes, exhibits ADHD-like behaviors in mice

To further understand whether the role of SELENOT is specific in dopaminergic neurons, we additionally constructed *Selenot*<sup>fl/fl</sup>;*Nestin-cre* and *Selenot*<sup>fl/fl</sup>;*Gfap-cre* mice to knock out *Selenot* expression in the whole brain and astrocytes, respectively, and assessed ADHD-like behaviors in these mice (Appendix Figs. S3–S5). The knockouts were confirmed by Western blot and/or immunohistochemical analyses (Appendix Figs. S3A–C and S5A,B). Although *Selenot*<sup>fl/fl</sup>;*Gfap-cre* mice grew normally, body weight of *Selenot*<sup>fl/fl</sup>;*Nestin-cre* mice was lower than that of *Selenot*<sup>fl/fl</sup> mice at 8 weeks of age (Appendix Fig. S3D), despite no signs of premature cell death.

In open field test (Appendix Fig. S3E), *Selenot*<sup>fl/fl</sup>;*Nestin-cre* mice exhibited higher spontaneous locomotion (distance traveled every 10 min and total distance) and active time (Appendix Fig. S3F–H), but normal mean speed, center distance, and center time (Appendix Fig. S3I–K), compared to *Selenot*<sup>fl/fl</sup> mice. Interestingly, the hyperactivity appeared to be transient and disappeared after an hour (Appendix Fig. S3F). In the elevated plus maze test, *Selenot*<sup>fl/fl</sup>;*Nestin-cre* mice showed increased total entries to arms (Appendix Fig. S3L), but normal percentages of time spent in the open arms and entries to the open arms (Appendix Fig. S3M,N). These results suggest hyperlocomotion and no anxiety in *Selenot*<sup>fl/fl</sup>;*Nestin-cre* mice. The mice also displayed some repetitive behaviors, such as in rearing and digging, but not others, including tail flicking, self-grooming, climbing, and jumping (Appendix Fig. S3O). In contrast, *Selenot*<sup>fl/fl</sup>;*Nestin-cre* mice displayed no impairment in 1) motor coordination, as indicated by the latency to fall in the rotarod test (Appendix Fig. S4A,B), 2) short-term and long-term recognition memory, as suggested by novel object recognition tests (Appendix Fig. S4C–H), and 3) social interaction, including social preference (Appendix Fig. S4I–K) and social recognition (Appendix Fig. S4L–N), based on the three-chamber sociability test. Taken together, these results suggest that the neurological consequences of

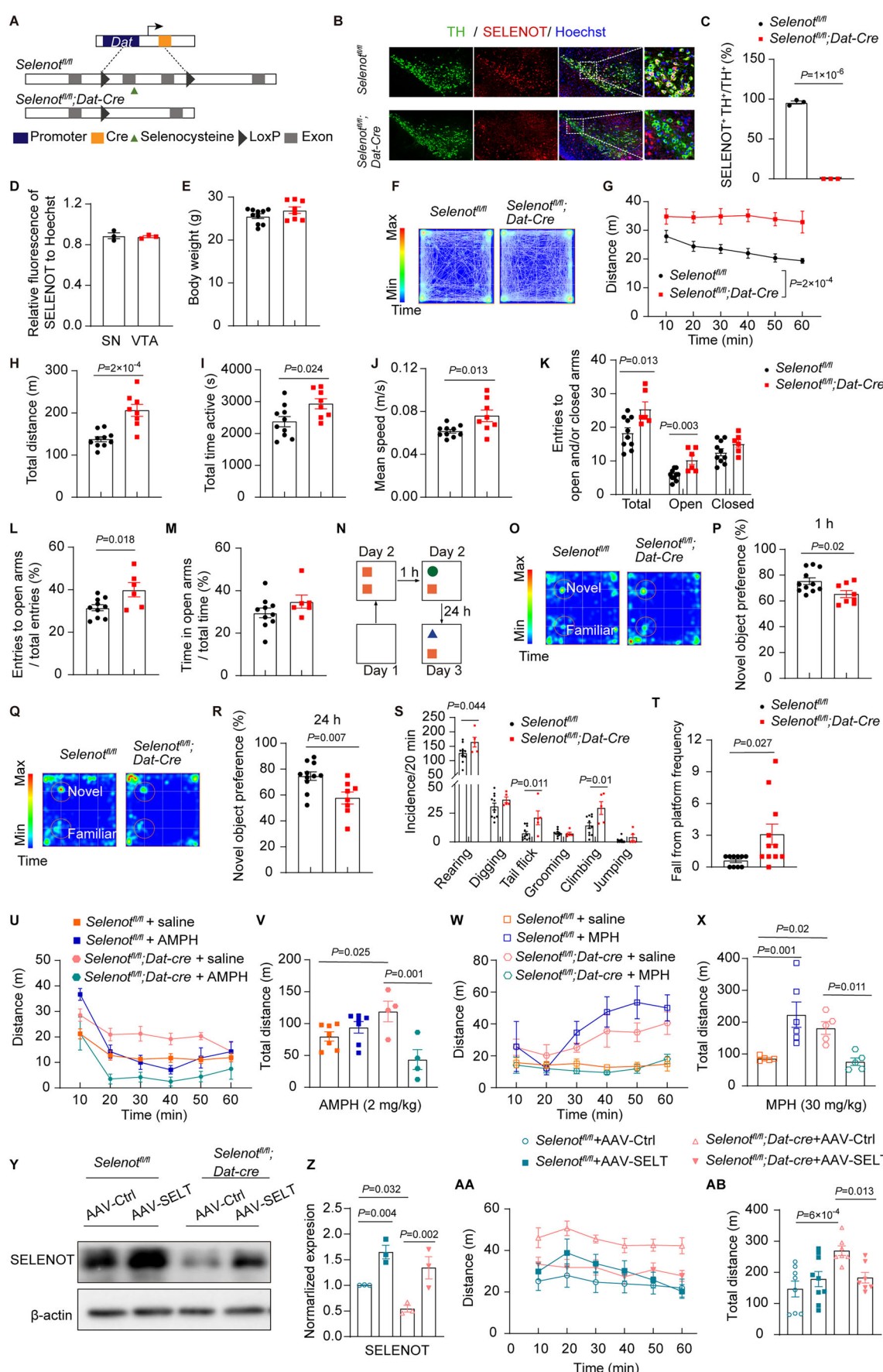

**Figure 1. *Selenot^{fl/fl};Dat-cre* mice exhibit hyperlocomotion and impaired memory.**

(A) Schematic diagram of *Dat* promoter-driven excision of *Selenot* exons 2 and 3. (B–D) Immunofluorescence analyses of SELENOT expression in dopaminergic neurons in *Selenot^{fl/fl}* (n = 3) and *Selenot^{fl/fl};Dat-cre* (n = 3) mice. Presented are the representative images (B), the percentage of SELENOT and TH colocalized cells relative to TH-positive cells (C), and relative fluorescence intensity of SELENOT to the nuclei staining in the SN and VTA (D). Green, TH; red, SELENOT; blue, Hoechst. (E) Body weight of *Selenot^{fl/fl}* (n = 10) and *Selenot^{fl/fl};Dat-cre* (n = 8) mice at 8 weeks of age. (F–J) Open field test of *Selenot^{fl/fl}* (n = 10) and *Selenot^{fl/fl};Dat-cre* (n = 8) mice. Presented are representative activity heatmap (F), distance traveled every 10 min (G), total distance traveled (H), total active time (I), and mean speed (J). (K–M) Elevated plus maze test of *Selenot^{fl/fl}* (n = 10) and *Selenot^{fl/fl};Dat-cre* (n = 6) mice. Presented are entries to total, open, and closed arms (K), percentage of entries to open arms compared to total entries (L), and percentage of time spent in open arms to total time spent in open and closed arms (M). (N–R) Novel object recognition test of *Selenot^{fl/fl}* (n = 11) and *Selenot^{fl/fl};Dat-cre* (n = 8) mice. Presented are schematic diagram (N), representative activity heatmap, and novel object preference in short-term memory (1 h; O, P) and long-term memory (24 h; Q, R). (S) Stereotyped behaviors of *Selenot^{fl/fl}* (n = 11) and *Selenot^{fl/fl};Dat-cre* (n = 5) mice. (T) Fall from platform frequencies of *Selenot^{fl/fl}* (n = 10) and *Selenot^{fl/fl};Dat-cre* (n = 11) mice in the cliff avoidance reaction test. (U, V) Open field test of mice treated with saline or 2 mg/kg AMPH. n = 7 *Selenot^{fl/fl}* + saline mice, 7 *Selenot^{fl/fl}* + AMPH mice, 4 *Selenot^{fl/fl};Dat-cre* + saline mice, and 4 *Selenot^{fl/fl};Dat-cre* + AMPH mice. (W, X) Open field test of mice treated with saline or 30 mg/kg MPH. n = 5 *Selenot^{fl/fl}* + saline mice, 6 *Selenot^{fl/fl}* + MPH mice, 5 *Selenot^{fl/fl};Dat-cre* + saline mice, and 5 *Selenot^{fl/fl};Dat-cre* + MPH mice. Presented are locomotor activity every 10 min (U, W) and total distance for 60 min (V, X). (Y, Z) Western blot analyses (Y) and quantifications (Z) of SELENOT in the substantia nigra of mice infected with AAVs. n = 3 mice for each group. Quantifications are normalized to β-actin. (AA, AB) Open field test of mice infected with AAVs. n = 8 *Selenot^{fl/fl}* + AAV-Ctrl mice, 9 *Selenot^{fl/fl}* + AAV-SELT mice, 7 *Selenot^{fl/fl};Dat-cre* + AAV-Ctrl mice, and 7 *Selenot^{fl/fl};Dat-cre* + AAV-SELT mice. Presented are locomotor activity every 10 min (AA) and total distance for 60 min (AB). Data are presented as means ± SEM and analyzed by two-way repeated measures ANOVA for (G), factorial ANOVA for (V, X, Z, and AB), and two-tailed unpaired *t*-test for other comparisons. AAV adeno-associated virus; AAV-Ctrl control AAV AAV-SELT SELENOT-expressing AAV, AMPH amphetamine, MPH methylphenidate, SN substantia nigra, TH tyrosine hydroxylase, VTA ventral tegmental area. Source data are available online for this figure.

*Selenot* deficiency in the whole brain are reminiscent of those in dopaminergic neurons, displaying ADHD-like behaviors.

Next, *Selenot^{fl/fl};Gfap-cre* mice were assessed using open field and repetitive behavior tests. Compared to *Selenot^{fl/fl}* mice, *Selenot^{fl/fl};Gfap-cre* mice showed no signs of 1) hyperactivity, as indicated by locomotion, active time, and mean speed in the open field test (Appendix Fig. S5C–G), 2) anxiety, as evidenced by normal percentages of center distance and center time in the open field test (Appendix Fig. S5H, I), and 3) repetitive behaviors, as indicated by results for rearing, tail flicking, self-grooming, climbing, and jumping (Appendix Fig. S5J). Interestingly, the digging frequency was slightly decreased ($P < 0.05$) in the *Selenot^{fl/fl};Gfap-cre* mice (Appendix Fig. S5J). These results suggest that *Selenot* deficiency in astrocytes does not lead to ADHD-like phenotypes.

## Impaired dopamine metabolism and reuptake and enhanced postsynaptic excitatory input in the striatum of *Selenot^{fl/fl};Dat-cre* mice

To gain insight into the behavioral changes, key striatal neurotransmitters [norepinephrine, γ-aminobutyric acid (GABA), acetylcholine, 5-hydroxytryptamine, and dopamine (DA)] and DA metabolites [3,4-dihydroxyphenylacetic acid (DOPAC), homovanillic acid (HVA), and 3-methoxytyramine (3-MT)] were assessed in the whole striatum. Results showed that the levels of these chemicals were not markedly different between *Selenot^{fl/fl};Dat-cre* and *Selenot^{fl/fl}* mice (Fig. 2A–E). However, the ratios of 3-MT/DA and HVA/DA were significantly increased in *Selenot^{fl/fl};Dat-cre* mice (Fig. 2F), suggesting accelerated dopamine metabolism. Next, results using striatal microdialyzed fluids from live mice (Fig. 2G) showed that the extrasynaptic dopamine level was significantly higher in *Selenot^{fl/fl};Dat-cre* mice than in *Selenot^{fl/fl}* mice (Fig. 2H). These results indicate increased dopamine metabolism and a greater basal dopamine tone in the striatum of *Selenot^{fl/fl};Dat-cre* mice. In addition, levels of glutamic acid (Fig. 2I), but not 5-hydroxyindoleacetic acid (Fig. 2J; a 5-hydroxytryptamine metabolite), were increased in the striatum of *Selenot^{fl/fl};Dat-cre* mice.

We next determined whether *Selenot* deficiency affects neural activity using whole-cell electrophysiological recordings.

Dopaminergic neurons were identified by the co-immunostaining of biocytin and TH (Fig. 3A). The threshold and amplitude of action potentials remained unchanged in the substantia nigra of *Selenot^{fl/fl};Dat-cre* mice (Fig. 3B–D). The pacemaker frequency was also unchanged in the mice (Appendix Fig. S6A,B). Analyses of spontaneous excitatory postsynaptic currents (sEPSC) showed that the frequency, but not the amplitude, in medium spiny neurons was significantly enhanced in the striatum of *Selenot^{fl/fl};Dat-cre* mice (Fig. 3E–G). This result suggests enhanced postsynaptic excitatory input and is consistent with the increased abundance of extrasynaptic dopamine in the striatum of *Selenot^{fl/fl};Dat-cre* mice. Further electroencephalogram (EEG) analyses (Fig. 3H–J) showed that theta (3–8 Hz) power was significantly elevated (Fig. 3K), while EEG power at higher frequencies (> 8 Hz) appeared normal (Appendix Fig. S6C, D) in occipital regions of *Selenot^{fl/fl};Dat-cre* mice.

## *Selenot* deficiency leads to markedly reduced DAT expression and mildly elevated MAO-A expression, but no induction of midbrain cellular stress

To better understand why the striatal extrasynaptic dopamine level was elevated, we examined the levels of proteins in association with dopamine synthesis (i.e., TH), signaling [i.e., vesicular monoamine transporter 2 (VMAT2) and D2R; we could not find a specific antibody for D1R], reuptake (i.e., DAT), and metabolism [i.e., catechol-O-methyltransferase (COMT), monoamine oxidase A (MAO-A), and MAO-B] by Western blot analyses. Results showed that the DAT level was greatly reduced (by 57% and 79%, respectively), while MAO-A was elevated in both the dorsal and ventral striatum of *Selenot^{fl/fl};Dat-cre* mice. Levels of other proteins, including TH, were comparable between *Selenot^{fl/fl};Dat-cre* and *Selenot^{fl/fl}* mice (Fig. 4A,B). These findings regarding DAT and TH were corroborated by striatal immunohistochemical analyses (Fig. 4C–F). In line with the striatum findings, DAT levels were drastically reduced in the substantia nigra (by 61%) and the VTA (by 85%) of *Selenot^{fl/fl};Dat-cre* mice; interestingly, TH levels were mildly reduced in these two regions, as suggested by the Western

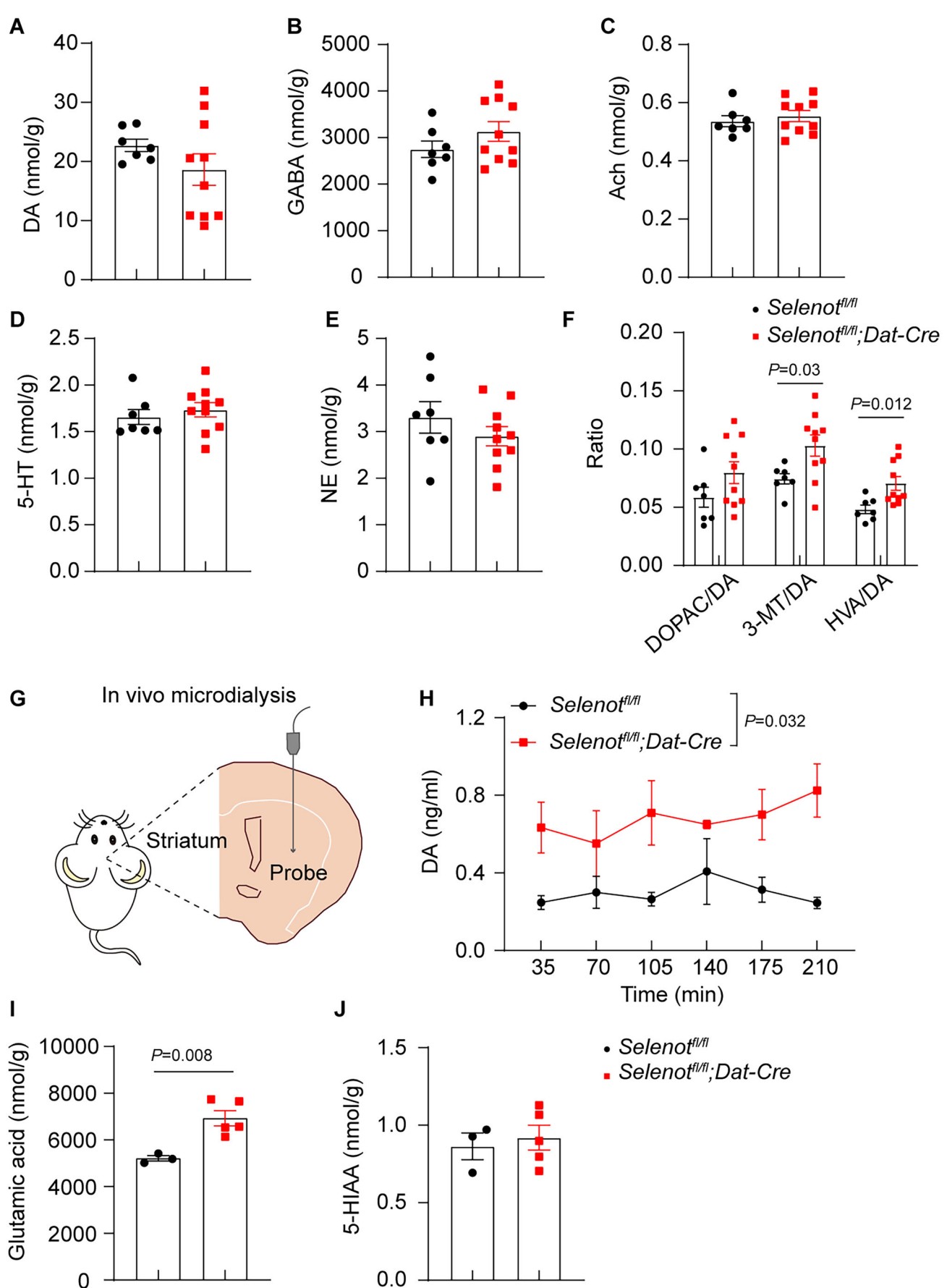

---

**Figure 2.   Impaired DA metabolism and reuptake in *Selenot^{fl/fl};Dat-cre* mice.**

(A–E) HPLC analyses of striatal neurotransmitters, including DA (A), GABA (B), Ach (C), 5-HT (D), and NE (E). (F) Ratio of DA metabolites to DA. $n = 7$ *Selenot^{fl/fl}* mice and 10 *Selenot^{fl/fl};Dat-cre* mice. (G, H) DA levels measured in the striatum by microdialysis in freely moving *Selenot^{fl/fl}* ($n = 3$) and *Selenot^{fl/fl};Dat-cre* ($n = 3$) mice. Presented are schematic diagram (G) and HPLC-analyzed DA levels (H). (I, J) HPLC analyses of striatal glutamate (I) and 5-HIAA (J) levels. Data are presented as means ± SEM and analyzed by two-way repeated measures ANOVA for (H) and two-tailed unpaired $t$-test for (A–F, I, J). 3-MT, 3-methoxytyramine; 5-HT, 5-hydroxytryptamine; 5-HIAA, 5-hydroxyindoleacetic acid; Ach, acetylcholine; DA, dopamine; DOPAC, 3,4-dihydroxyphenylacetic acid; GABA, γ-aminobutyric acid; HVA, homovanillic acid; NE, norepinephrine. Source data are available online for this figure.

---

blot analyses (Fig. 4G,H). Further TH immunostaining analyses demonstrated that the number of dopaminergic neurons was mildly reduced in the midbrain of *Selenot^{fl/fl};Dat-cre* mice (Fig. 4I,J), but cellular TH expression remained unchanged in the dopaminergic neurons of this genotype (Fig. 4K,L). These results suggest that the mild reduction of TH levels observed in the Western blot analyses was due to a slight reduction in dopaminergic cell number, rather than a change in intracellular TH expression.

We then assessed DAT and TH protein expression in *Selenot^{fl/fl}; Nestin-cre* and *Selenot^{fl/fl};Gfap-cre* mice in comparison to *Selenot^{fl/fl}* mice. Similar to *Selenot^{fl/fl};Dat-cre* mice, *Selenot^{fl/fl};Nestin-cre* mice also exhibited a remarkable reduction in DAT levels in the whole striatum and substantia nigra, along with a slight reduction in TH levels ($P = 0.074$) in the substantia nigra, but not in the striatum (Appendix Fig. S7A,B). In contrast, *Selenot^{fl/fl};Gfap-cre* mice displayed no changes in DAT and TH protein expression in the striatum and substantia nigra based on Western blot analyses (Appendix Fig. S7C,D). In addition, DAT expression did not differ in the whole striatum or substantia nigra between *Dat-cre* mice and *Selenot^{fl/fl}* mice (Appendix Fig. S8A,B), which was consistent with the description from Jackson Laboratory, where the *Dat-cre* line originated, and precluded potential interference of *Dat-cre* on DAT expression. Indeed, results from the open-field test still suggested that *Selenot^{fl/fl};Dat-cre* mice were hyperactive compared to the *Dat-cre* control mice (Appendix Fig. S8C–E). The substantia nigra of *Selenot^{fl/fl};Dat-cre* mice were not prone to apoptosis, as determined by Western blot analyses of caspase-3, cleaved caspase-3, and B-cell lymphoma-2 (BCL-2), and did not display ER stress, as indicated by analyses of binding-immunoglobulin protein (BIP) and CCAAT-enhancer-binding protein homologous protein (CHOP) (Appendix Fig. S8F,G).

Together, these results indicate in *Selenot^{fl/fl};Dat-cre* mice that 1) accelerated dopamine metabolism is likely associated with enhanced MAO-A expression; 2) the elevation of extrasynaptic dopamine levels corresponds with the reduction in DAT and the subsequent impairment in dopamine reuptake; 3) although the number of dopaminergic neurons slightly decreases in the midbrain, the striatal innervations are not significantly affected; 4) the decreased number of dopaminergic neurons is not due to cellular stress but is likely the result of proliferation arrest of progenitors during embryonic development, given that SELENOT can promote G1-to-S cell cycle transition (Shao et al, 2019); 5) similar to behaviors, comparable DAT and TH changes are found in *Selenot^{fl/fl};Nestin-cre* mice, but not in *Selenot^{fl/fl};Gfap-cre* mice.

## SELENOT directly regulates DAT expression through the transcription factor NURR1

RNA-seq was first performed using the substantia nigra of *Selenot^{fl/fl}* and *Selenot^{fl/fl};Dat-cre* mice to explore the mechanisms underlying the relationship between SELENOT and DAT. The number of valid bases after quality control in each sample exceeded 6.4 G. Both Q20 and Q30 values for each sample were over 98%, and no GC bias was found (Appendix Table S1). RNA-seq results revealed a total of 320 differentially expressed genes (DEGs), including 156 upregulated and 164 downregulated (Fig. 5A and Dataset EV1); however, this list did not include the 24 selenoprotein genes (Appendix Fig. S9A). Gene Ontology analysis of the DEGs highlighted biological processes such as dopamine biosynthesis, regulation of dopamine metabolism, and neurotransmitter secretion. Additionally, significant enrichments were observed in Ca²⁺-associated events, including calcium-dependent phospholipid binding, calcium ion binding, negative regulation of calcium ion transport, calcium-dependent protein kinase regulator activity, and calcium-dependent activation of synaptic vesicle fusion (Fig. 5B). Among the dopaminergic signaling-related genes, *Dat/Slc6a3* and *Nurr1/Nr4a2*, but not others, were found to be apparently downregulated in the substantia nigra of *Selenot^{fl/fl};Dat-cre* mice (Fig. 5A,C). Expression levels of the ER-associated calcium channel genes, including *Serca1/2/3* (also named as *Atp2a1/2/3*), *Ip3r1/2/3*, and *Ryr1/2/3*, were not significantly different between *Selenot^{fl/fl};Dat-cre* and *Selenot^{fl/fl}* mice. Interestingly, *Serca2*, but not *Serca1* and *Serca3*, was abundantly expressed in the substantia nigra, while *Ip3r1* and *Ryr2* were more abundant than their fellow isoforms (Fig. 5D). Western blot analyses for SERCA2, IP3R1, and RYR2 using *Selenot^{fl/fl};Nestin-cre* and *Selenot^{fl/fl}* cortices showed that *Selenot* deficiency did not alter their protein expression levels (Appendix Fig. S9B,C). *Nurr1/Nr4a2* was among the 12 ADHD-associated DEGs identified in the mice (Fig. 5E). The downregulation of *Dat* and *Nurr1* was validated by qPCR analysis using the substantia nigra of *Selenot^{fl/fl}* and *Selenot^{fl/fl};Dat-cre* mice (Fig. 5F). This transcriptional change of *Dat* was consistent with the earlier Western blot results. Finally, we examined *Nurr1* expression at the protein level, as it is a known transcription factor for DAT (Sacchetti et al, 2001). Consistent with the transcriptional findings, NURR1 protein expression was significantly reduced in the substantia nigra of *Selenot^{fl/fl};Dat-cre* mice (Fig. 5G,H).

We further explored the link between SELENOT and DAT. In agreement with the in vivo results, knockdown of *SELENOT* led to reduced DAT protein levels in HEK293 cells. Conversely, over-expression of *SELENOT* increased DAT protein levels (Fig. 6A,B). These SELENOT-mediated regulations of *DAT* expression were also observed at the transcriptional level (Fig. 6C). Results from *NURR1* siRNA knockdown validated that NURR1 could regulate the transcriptional level of *DAT* (Fig. 6D). Consistent with the *DAT* results, knockdown and overexpression of *SELENOT*, respectively, downregulated and upregulated the protein and mRNA expression of *NURR1* (Fig. 6E–G). We further demonstrated that siRNA

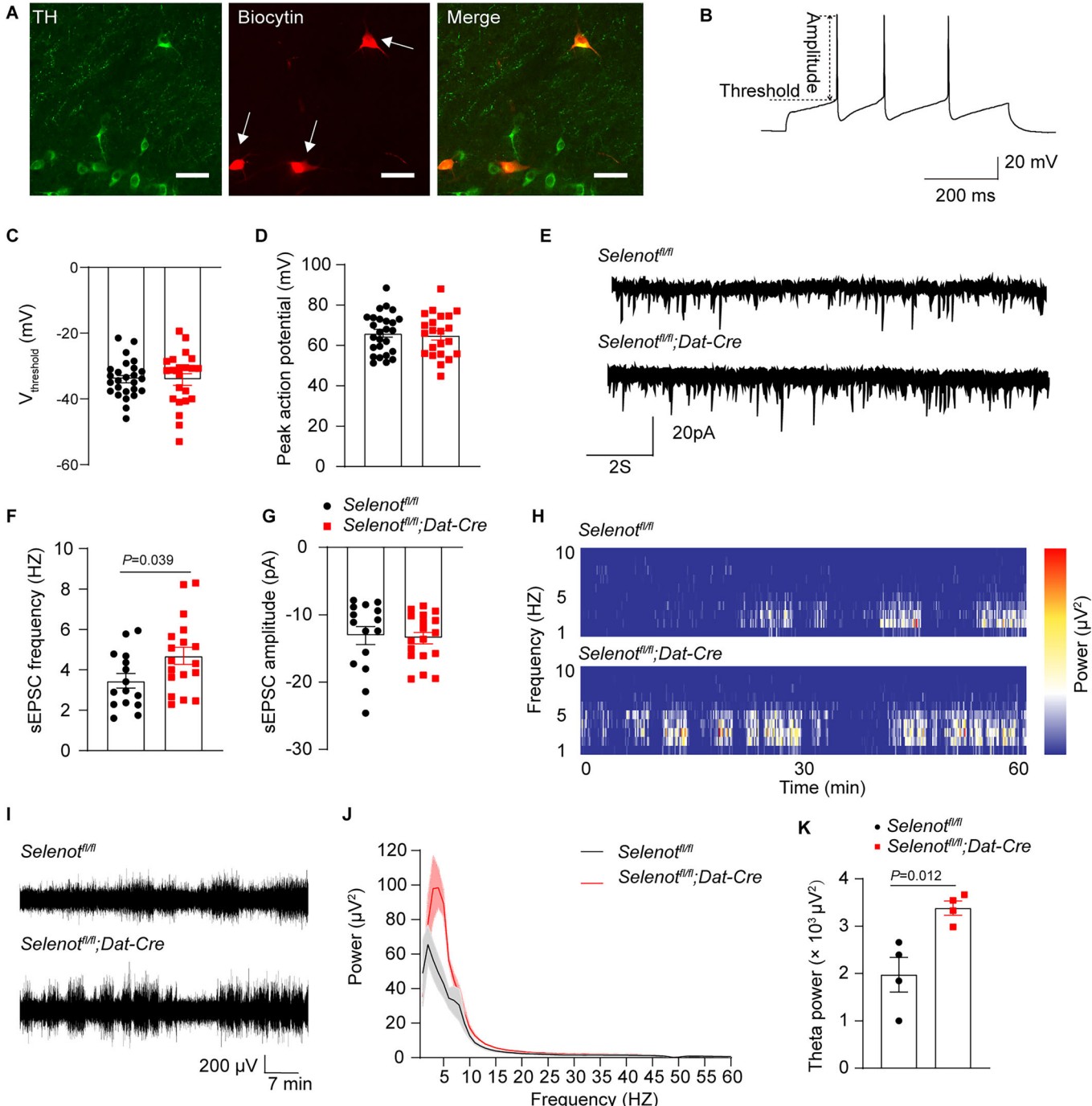

**Figure 3. *Selenot^{fl/fl};Dat-cre* mice display normal dopaminergic neuron firing but elevated postsynaptic excitatory input and EEG theta power.**

(A) Representative images showing that the recorded neurons (white arrows) were positive for TH (green) on a polyoxymethylene-fixed brain slice after patching. Biocytin (red) was diffused into cells by whole-cell patch clamp. Scale bar, 50 μm. (B–D) Action potential of dopaminergic neurons. Presented are representative action potential recorded under current clamp mode (B), and quantified thresholds (C) and amplitudes (D) of action potentials. $n = 26$ neurons recorded from 8 *Selenot^{fl/fl}* mice and 22 neurons recorded from 8 *Selenot^{fl/fl};Dat-cre* mice. (E–G) sEPSC in the striatum. Presented are representative recordings of sEPSC (E), and quantified frequencies (F) and amplitudes (G) of striatal medium spiny neurons. $n = 15$ neurons recorded from 4 *Selenot^{fl/fl}* mice and 18 neurons recorded from 3 *Selenot^{fl/fl};Dat-cre* mice. (H–K) EEG analysis. Presented are representative spectrogram (H) and traces (I), and mean EEG power spectral density (J) and theta power (K) over 60 min. $n = 4$ *Selenot^{fl/fl}* mice and 4 *Selenot^{fl/fl};Dat-cre* mice. Data are presented as means ± SEM and analyzed by two-tailed unpaired *t*-test. EEG, electroencephalogram; sEPSC, spontaneous excitatory postsynaptic currents; TH, tyrosine hydroxylase; V_{threshold}, threshold of action potential. Source data are available online for this figure.

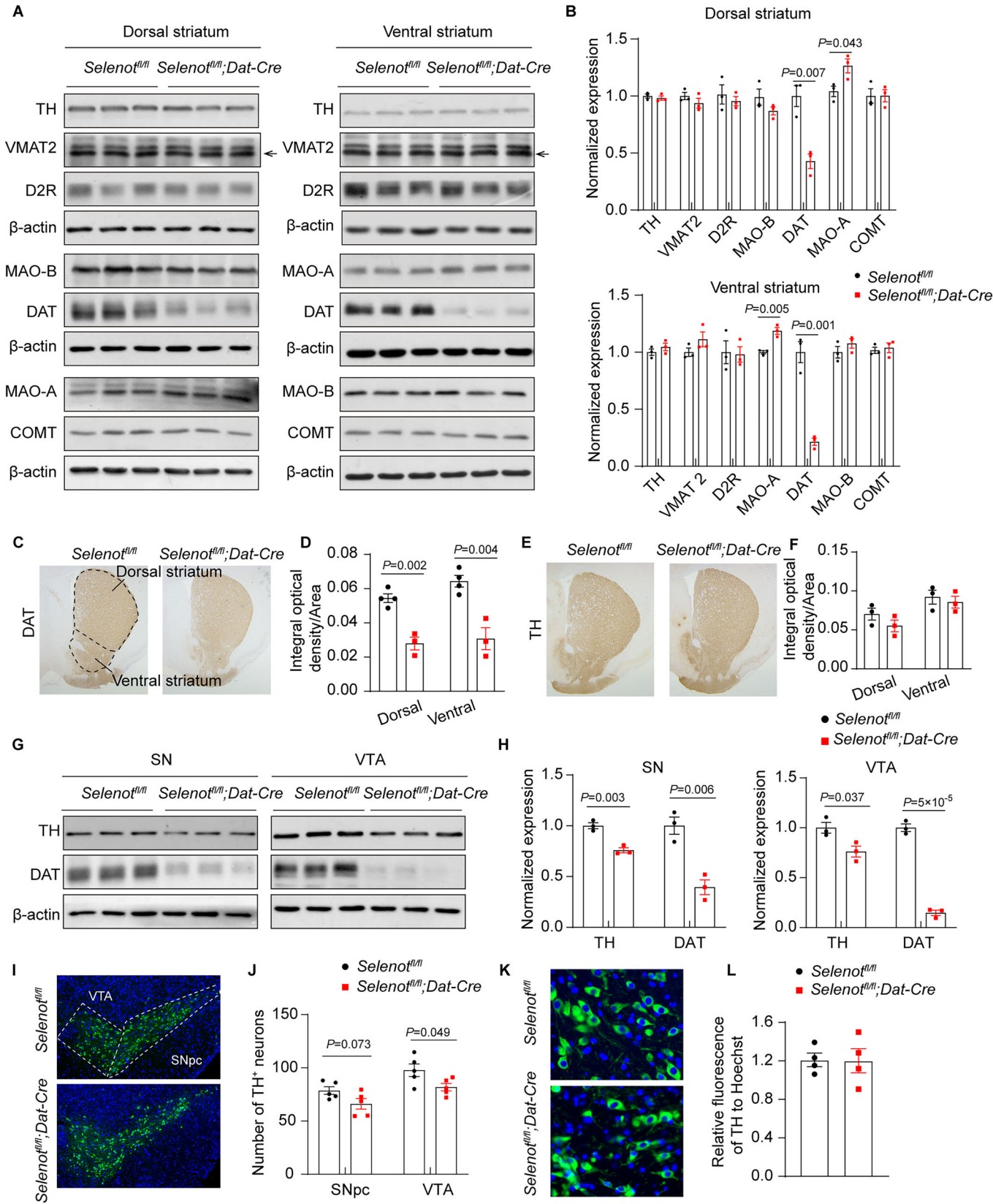

**Figure 4.   Altered expression of dopamine signaling-associated proteins in *Selenot^{fl/fl};Dat-cre* mice.**

(A, B) Western blot analyses (A) and quantifications (B) of proteins in the dorsal and ventral striatum. The order in (B) corresponds to that in (A). The arrow indicates the band for VMAT2. The β-actin controls were from the same or other gels run in parallel. $n = 3$ *Selenot^{fl/fl}* mice and 3 *Selenot^{fl/fl};Dat-cre* mice. (C–F) Immunohistochemical analyses (C, E) and optical density quantifications (D, F) of DAT and TH in the striatum. $n = 3$–4 *Selenot^{fl/fl}* mice and 3 *Selenot^{fl/fl};Dat-cre* mice. (G, H) Western blot analyses (G) and quantifications (H) of TH and DAT in the midbrain. The β-actin controls were from the same or other gels run in parallel. $n = 3$ *Selenot^{fl/fl}* mice and 3 *Selenot^{fl/fl};Dat-cre* mice. (I, J) Immunofluorescence (I) and counts (J) of dopaminergic neurons in the midbrain. $n = 5$ *Selenot^{fl/fl}* mice and 5 *Selenot^{fl/fl};Dat-cre* mice. (K, L) Immunofluorescence (K) and cellular quantifications (L) of TH expression in dopaminergic neurons. $n = 4$ *Selenot^{fl/fl}* mice and 4 *Selenot^{fl/fl};Dat-cre* mice. Quantifications are normalized to β-actin for Western blot. Each immunostaining quantification was averaged from 2 to 3 consecutive slices. Data are presented as means ± SEM and analyzed by two-tailed unpaired *t*-test. COMT catechol-O-methyltransferase, DAT dopamine transporter; D2R dopamine D2 receptor, MAO-A monoamine oxidase A, MAO-B monoamine oxidase B, SN substantia nigra, SNpc substantia nigra pars compacta, TH tyrosine hydroxylase, VMAT2 vesicular monoamine transporter 2, VTA ventral tegmental area. Source data are available online for this figure.

knockdown of *NURR1* expression blunted the SELENOT overexpression-induced upregulation of *DAT* level in HEK293 cells (Fig. 6H). Furthermore, viral-mediated overexpression of SELENOT in dopaminergic neurons upregulated NURR1 and DAT expression in the substantia nigra of both *Selenot^{fl/fl}* and *Selenot^{fl/fl};Dat-cre* mice (Fig. 6I,J). Accordingly, this overexpression rescued the reduced protein levels of DAT in the striatum of *Selenot^{fl/fl};Dat-cre* mice (Fig. 6K,L).

## SELENOT modulates ER-cytosol Ca²⁺ flux via the SERCA pump, thereby regulating NURR1 and DAT expression

We thus explored how SELENOT modulates $Ca^{2+}$ signaling. *SELENOT* siRNA knockdown reduced steady-state cytosolic $Ca^{2+}$ levels in HEK293 cells (Fig. 7A,B). Three $Ca^{2+}$ channels are known to be located on the ER membrane: sarco-ER $Ca^{2+}$ ATPase (SERCA), ryanodine receptor (RYR), and inositol 1,4,5 trisphosphate receptor (IP3R). Among them, SERCA mediates $Ca^{2+}$ influx into the ER, whereas RYR and IP3R mediate $Ca^{2+}$ efflux to the cytosol (Bagur and Hajnoczky, 2017). Next, three available chemical interferers for these channels were employed: cyclopiazonic acid (CPA; a SERCA antagonist), caffeine (a RYR agonist), and adenophostin A (AdA; an IP3R agonist). *SELENOT* siRNA knockdown in HEK293 cells reduced the peak increment of cytosolic $Ca^{2+}$ concentration when treated with CPA (15 µM; Fig. 7C), but not when treated with caffeine or AdA (10 mM and 1 µM, respectively; Fig. 7D,E), Conversely, *SELENOT* overexpression increased steady-state cytosolic $Ca^{2+}$ levels (Fig. 7F,G) and enhanced the peak increment in response to CPA treatment (15 µM; Fig. 7H). Further analyses demonstrated that CPA treatment time- and dose-dependently upregulated the expression levels of *NURR1* and *DAT* (Fig. 7I,J), while *SELENOT* pre-knockdown blunted the CPA-stimulated *DAT* and *NURR1* expression (Fig. 7K). Notably, these effects should consider the partial contribution from the *SELENOT* knockdown per se. CPA treatment also appeared to stimulate *SELENOT* expression, likely due to a feedback response. Likewise, results from $Ca^{2+}$ imaging using AAV-GCaMP-infected brain slices (Fig. 7L) showed a reduction in steady-state cytosolic $Ca^{2+}$ levels (Fig. 7M) and a suppression of the peak increment in response to CPA treatment (30 µM; Fig. 7N) in the *Selenot^{fl/fl};Dat-cre* slices compared to *Dat-cre* controls. Together, these results suggest that SELENOT increases cytosolic $Ca^{2+}$ levels via inhibiting SERCA activity, and that the increase in cytosolic $Ca^{2+}$ promotes *NURR1* and *DAT* expression.

## SELENOT physically interacts with and colocalizes to SERCA2 in the ER

We thus performed co-immunoprecipitation coupled with mass spectrometry to identify potential SELENOT-interacting proteins, using SELENOT^{U49C} mutant to achieve higher expression levels as the bait protein (Fig. 8A). The assay identified 242 unique potential SELENOT^{U49C}-interacting proteins with protein scores over 30. Among the isoforms of three ER channels, SERCA, IP3R and RYR, we found that SERCA2 was significantly enriched, with 41 peptides and protein score of 188, while none of the others qualified (Fig. 8B,C). In addition, the SELENOT-interacting proteins were found to be enriched in a number of human phenotypes and diseases, including hyperactivity, neurodevelopmental abnormality, and central nervous system disease (Fig. 8D).

We next performed an independent co-immunoprecipitation validation using SELENOT^{U49C} as the bait. The results showed that SELENOT^{U49C} could indeed pull down SERCA2 and also appeared to interact with IP3R1, but not RYR2 (Fig. 8E,F). However, a reverse co-immunoprecipitation assay showed that SELENOT^{U49C} could be pulled down by SERCA2 (Fig. 8G,H), but not by IP3R1 (Fig. 8I,J). Further immunofluorescence colocalization analyses confirmed that SERCA2 colocalized with both SELENOT^{U49C} and wild-type SELENOT in the ER (Fig. 8K–N).

## Discussion

Significant progress has been made in understanding the etiology, lesioned processes and systems, and treatments of ADHD. However, some uncertainties remain, such as how to address heterogeneity for precision treatment, develop new accepted working models, and identify specific targets for long-term efficacious interventions (Posner et al, 2020). In the present study, we demonstrate that *Selenot* deficiency in dopaminergic neurons leads to ADHD-like behaviors in mice. The hyperactivity is ameliorated by psychostimulant treatment or SELENOT overexpression. The underlying pivotal mechanisms driving the phenotype involve the release of SELENOT-inhibited SERCA2 pump activity on the ER membrane, which subsequently reduces cytosolic $Ca^{2+}$ levels, thereby abolishing NURR1 and DAT expression. The DAT defect impairs presynaptic dopamine reuptake, leading to enhanced extrasynaptic dopamine retention and increased postsynaptic excitatory input in the striatum (Fig. 9).

A few rodent models of ADHD targeting different genes have been reported, each exhibiting differential relevant symptoms

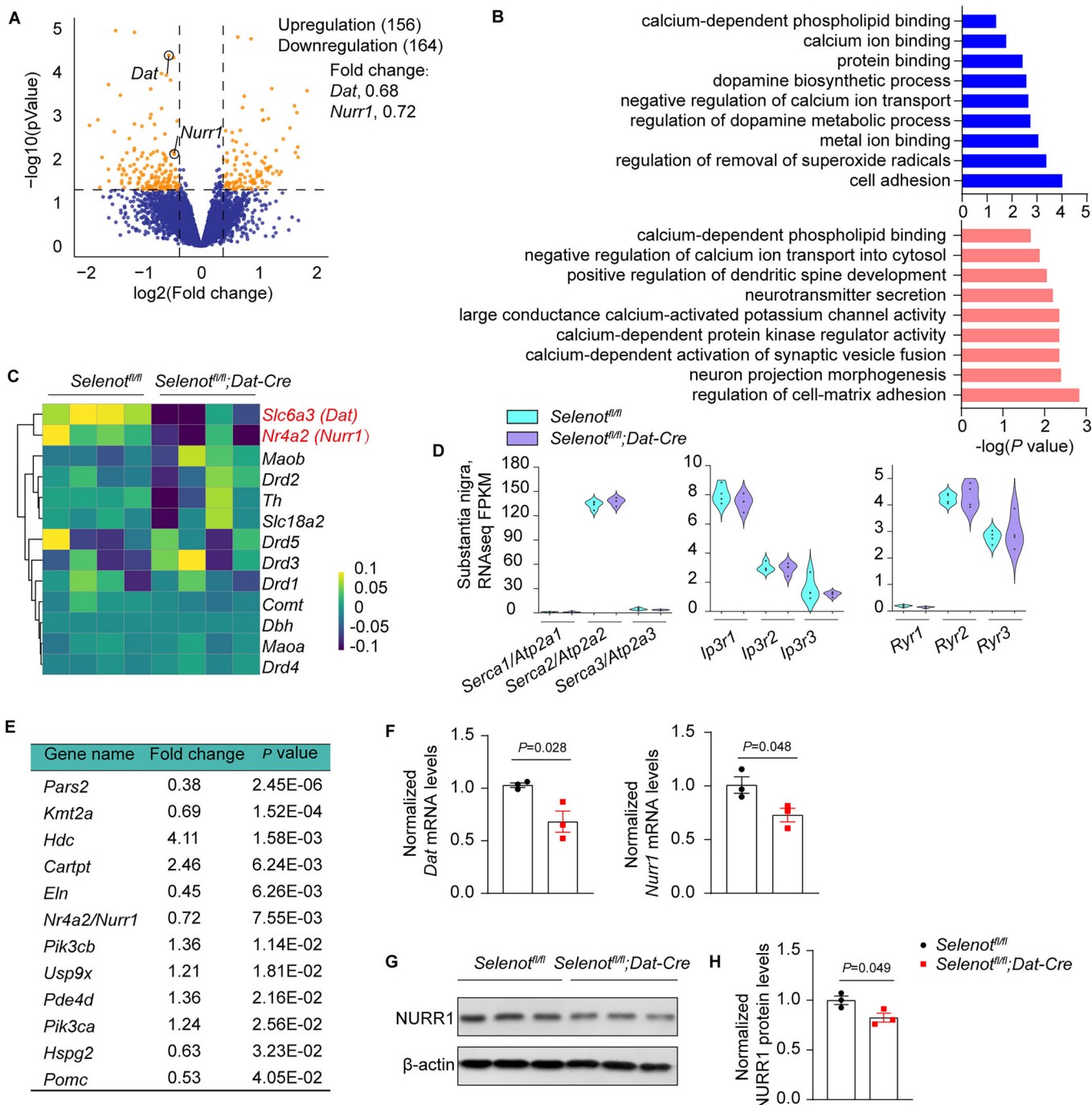

**Figure 5. RNA sequencing analysis of the substantia nigra in *Selenot^fl/fl^;Dat-cre* mice comparing to *Selenot^fl/fl^* mice.**

(**A**) Volcano plot of DEGs. Fold change >1.2, *P* < 0.05. Data were analyzed by Wald test. (**B**) Gene Ontology analysis for the DEGs. Data were analyzed by hypergeometric test. Blue, downregulated; Pink, upregulated. (**C**) Row-normalized RNA-seq heatmap for dopaminergic signaling-related genes. (**D**) Violin plot of RNA-seq FPKMs for *Atp2a/Serca*, *Ip3r*, and *Ryr* isoforms. (**E**) List of ADHD candidate DEGs identified by DisGenet. This database contains a total of 842 genes. (**F**) qPCR validation of *Dat* and *Nurr1* mRNA expression in the substantia nigra. (**G**, **H**) Western blot analysis (**G**) and quantifications (**H**) of NURR1 protein levels in the substantia nigra. *n* = 3 *Selenot^fl/fl^* mice and 3 *Selenot^fl/fl^;Dat-cre* mice. Quantifications are normalized to β-actin for qPCR and Western blot. Data are presented as means ± SEM and analyzed by two-tailed unpaired *t*-test for (**F**, **H**). DEG, differentially expressed gene; FPKM, fragments per kilobase per million. Source data are available online for this figure.

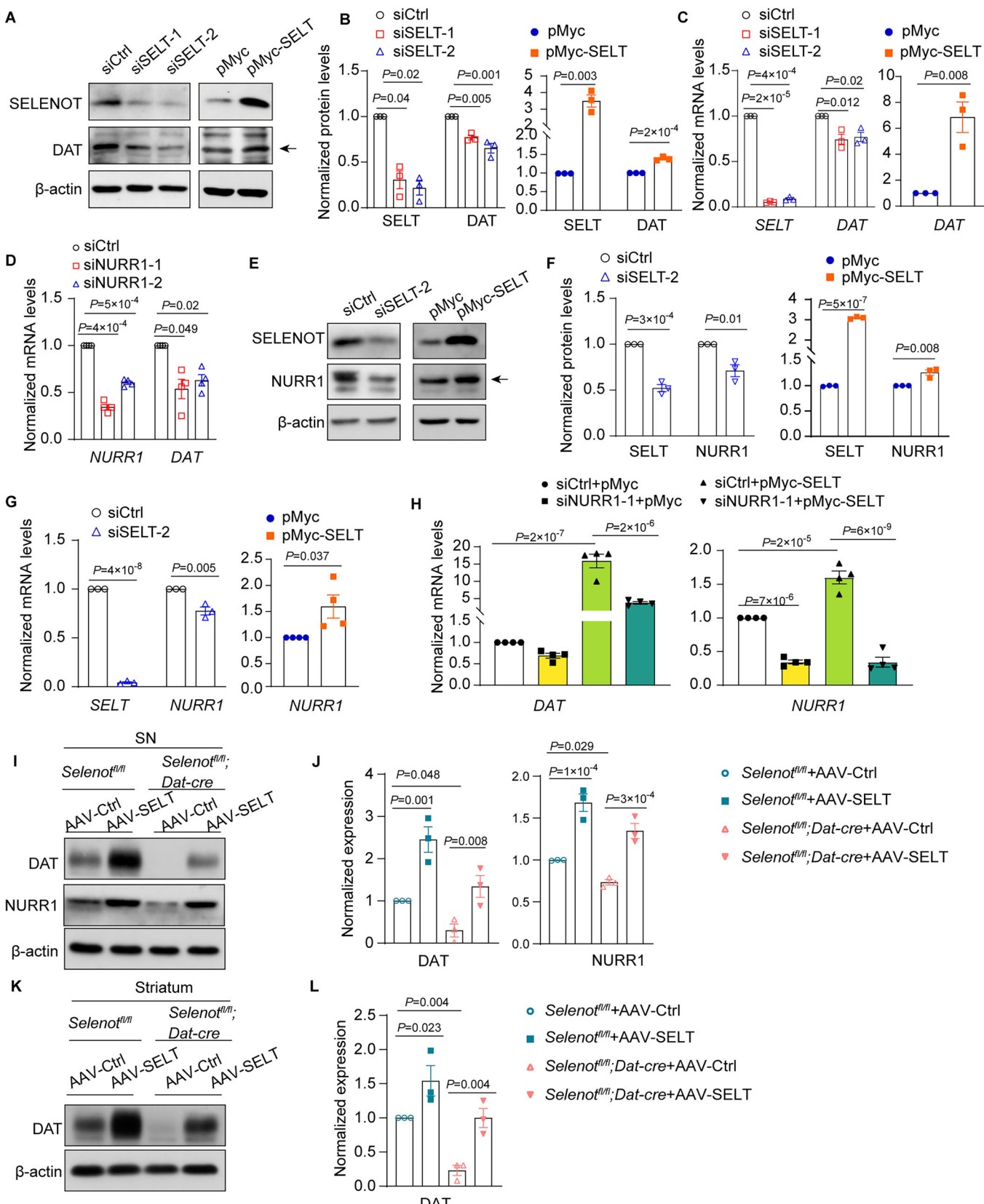

**Figure 6. Regulation of SELENOT on DAT and NURR1 expression.**

(A–C) Western blot analyses (A) and quantifications (B), and qPCR analyses (C) for DAT or SELENOT protein and mRNA levels in HEK293 cells transfected with siSELT-1, siSELT-2, siCtrl, pMyc-SELT, or pMyc (48 h for qPCR and 72 h for Western blot). The arrow indicates the band for DAT. $n = 3$ independent experiments. (D) Validation of the transcriptional regulation of DAT expression by NURR1. HEK293 cells were transfected with siNURR1-1, siNURR1-2 or siCtrl for 48 h. $n = 4$ independent experiments. (E–G) Western blot analyses (E) and quantifications (F), and qPCR analyses (G) for NURR1 protein and mRNA levels in HEK293 cells transfected with siSELT-2, siCtrl, pMyc-SELT, or pMyc (48 h for qPCR and 72 h for Western blot). The arrow indicates the band for NURR1. $n = 3$-4 independent experiments. (H) qPCR analyses for the effect of NURR1 on the SELENOT-induced *DAT* expression. Cells were pre-transfected with siNURR1 or siCtrl for 24 h, followed by transfection with pMyc-SELT or pMyc for 48 h. $n = 4$ independent experiments. Quantifications are normalized to their respective controls after normalization to β-actin for qPCR and Western blot. (I, J) Western blot analyses (I) and quantifications (J) of DAT and NURR1 in the substantia nigra. The β-actin controls were from the same or other gels run in parallel. (K, L) Western blot analyses (K) and quantifications (L) of DAT in the whole striatum. The *Selenot^{fl/fl}* and *Selenot^{fl/fl};Dat-cre* mice were infected with AAV-Ctrl or AAV-SELT. $n = 3$ mice for each group. Quantifications are normalized to β-actin. Data are presented as means ± SEM and analyzed by one-way ANOVA followed by Dunnett's post hoc test for (left panels of B and C, and D), factorial ANOVA for (H, J, and L), and two-tailed unpaired *t*-test for other comparisons. AAV adeno-associated virus, AAV-Ctrl control AAV, AAV-SELT SELENOT-expressing AAV, DAT dopamine transporter, NURR1 nuclear receptor-related 1, pMyc pCMV-Myc empty vector, pMyc-SELT Myc-tagged SELENOT vector, siCtrl scramble siRNA, siNURR1 *NURR1* siRNA, siSELT *SELENOT* siRNA. Source data are available online for this figure.

(Gallo and Posner, 2016). For instance, high-impulsive rats show impulsive behavior and deficits in premature responding, with reduced DRD2 expression in the nucleus accumbens (Dalley et al, 2011; Dalley et al, 2007). G protein-coupled receptor kinase interacting protein-1 (*GIT1*) knockout mice display hyperactivity and impaired learning and memory, with enhanced cortical EEG theta rhythms, reduced brain RAC1 signaling, and decreased hippocampal inhibitory presynaptic input (Won et al, 2011). The earliest animal model is presumably the DAT knockout mouse, which exhibits hyperactivity and impaired cued and spatial learning in association with persistent extracellular hyperdopaminergic tone (Gainetdinov et al, 1999; Giros et al, 1996; Weiss et al, 2007). In the present study, the phenotypes of *Selenot^{fl/fl};Dat-cre* mice include hyperactivity, recognition memory deficits, repetitive movements, and impulsivity, and they are sensitive to amphetamine or methylphenidate treatment. However, no marked impairments are found in anxiety, spatial memory, motor coordination, or social interaction. The phenotypes of *Selenot^{fl/fl};Nestin-cre* mice mostly resemble those of *Selenot^{fl/fl};Dat-cre* mice, in particular the hyperactivity. In contrast, *Selenot^{fl/fl};Gfap-cre* mice do not exhibit any of these behaviors. These results suggest that the modulation of ADHD-like behaviors is primarily attributed to SELENOT function in dopaminergic neurons, but not in astrocytes. Indeed, the dopaminergic system is one of the key neurotransmission circuits disturbed in ADHD, particularly as manifested by the first-line drugs, AMPH and MPH, both of which target this system (Thapar and Cooper, 2016).

Previous models have suggested pathological defects in the monoaminergic system, synaptic transmission, cell adhesion, and signal transduction (Gallo and Posner, 2016). In the present study, Ca²⁺ is the regulatory target of SELENOT via SERCA2, and DAT principally links SELENOT to the ADHD-like phenotypes. Both *Selenot^{fl/fl};Dat-cre* mice and *Selenot^{fl/fl};Nestin-cre* mice exhibit a drastic reduction in DAT in the VTA, substantia nigra, and striatum. However, unlike *Selenot^{fl/fl};Dat-cre* mice, which grow normally, *Selenot^{fl/fl};Nestin-cre* mice are growth-retarded as evidenced by our data and the previous study (Castex et al, 2016), suggesting a role of SELENOT in development maintenance. In addition, we speculate that whole-brain *Selenot* deficiency leads to neurocircuit compensation in the brain, which may explain why the phenotypes are less severe in *Selenot^{fl/fl};Nestin-cre* mice than in *Selenot^{fl/fl};Dat-cre* mice. The regulation of DAT by SELENOT is corroborated by the in vitro and in vivo assays with expression

manipulations. We further identify SELENOT as a positive regulator of NURR1. This nuclear receptor not only transcriptionally activates DAT expression (Sacchetti et al, 2001), but is also involved in several brain disorders, such as Parkinson's disease (Willems and Merk, 2022). NURR1 deficiency has also been reported to be associated with ADHD-like phenotypes in mice, including hyperlocomotion and impulsivity, but not anxiety or alterations in motor coordination, sociability, and memory (Montarolo et al, 2019). As noted earlier, SELENOT is implicated in the modulation of Ca²⁺ levels (Grumolato et al, 2008; Shao et al, 2019). Here, we conclude that the SELENOT-mediated ER-cytosol Ca²⁺ flow occurs via its interaction with SERCA2, at least in dopaminergic neurons, inhibiting its pump activity, but not through the efflux channels, RYRs and IP3Rs. Nonetheless, the impact of Ca²⁺ on DAT was less understood previously. There was only one study, consistent with our findings, showing that DAT activity, as determined by dopamine uptake, is reduced upon treatment with nifedipine (an L-type Ca²⁺ channel blocker) in mesencephalic cultures (Padmanabhan et al, 2008). In contrast, the transcription factor NURR1 has been shown to be regulatable by nifedipine and L-type calcium channel agonists (Kovalovsky et al, 2002; Tokuoka et al, 2014), and we confirm that NURR1 is a target of Ca²⁺ signaling.

Our findings present a novel Ca²⁺- and dopamine-centered ADHD animal model. DAT knockout mice exhibit impaired growth, reduced survival rates, and premature death (Giros et al, 1996), which are not observed in *Selenot^{fl/fl};Dat-cre* mice. Among the behaviors, hyperactivity and impulsivity are characterized in both *Selenot^{fl/fl};Dat-cre* mice and DAT knockout mice (Gainetdinov et al, 1999; Mereu et al, 2017), but they differ in spatial learning and recognition memory. Therefore, the abolished DAT and diminished dopamine reuptake in presynaptic terminals appear to be key, but not the sole, pathological events leading to the phenotypes in *Selenot^{fl/fl};Dat-cre* mice. In addition, the SELENOT-induced upregulation of DAT expression rescues the hyperactivity observed in *Selenot^{fl/fl};Dat-cre* mice but has no impact on the locomotion of *Selenot^{fl/fl}* control mice. Indeed, basal locomotion in normal mice overexpressing DAT remains unchanged (Salahpour et al, 2008). *Selenot* deficiency does not cause damage to presynaptic neural firing, nor does it induce apoptotic or ER cellular stress. The observation that ER stress is unaffected contrasts with findings in corticotrope cells, where *Selenot* deficiency was shown to induce ER stress (Hamieh et al, 2017). Likely, a compensatory mechanism is

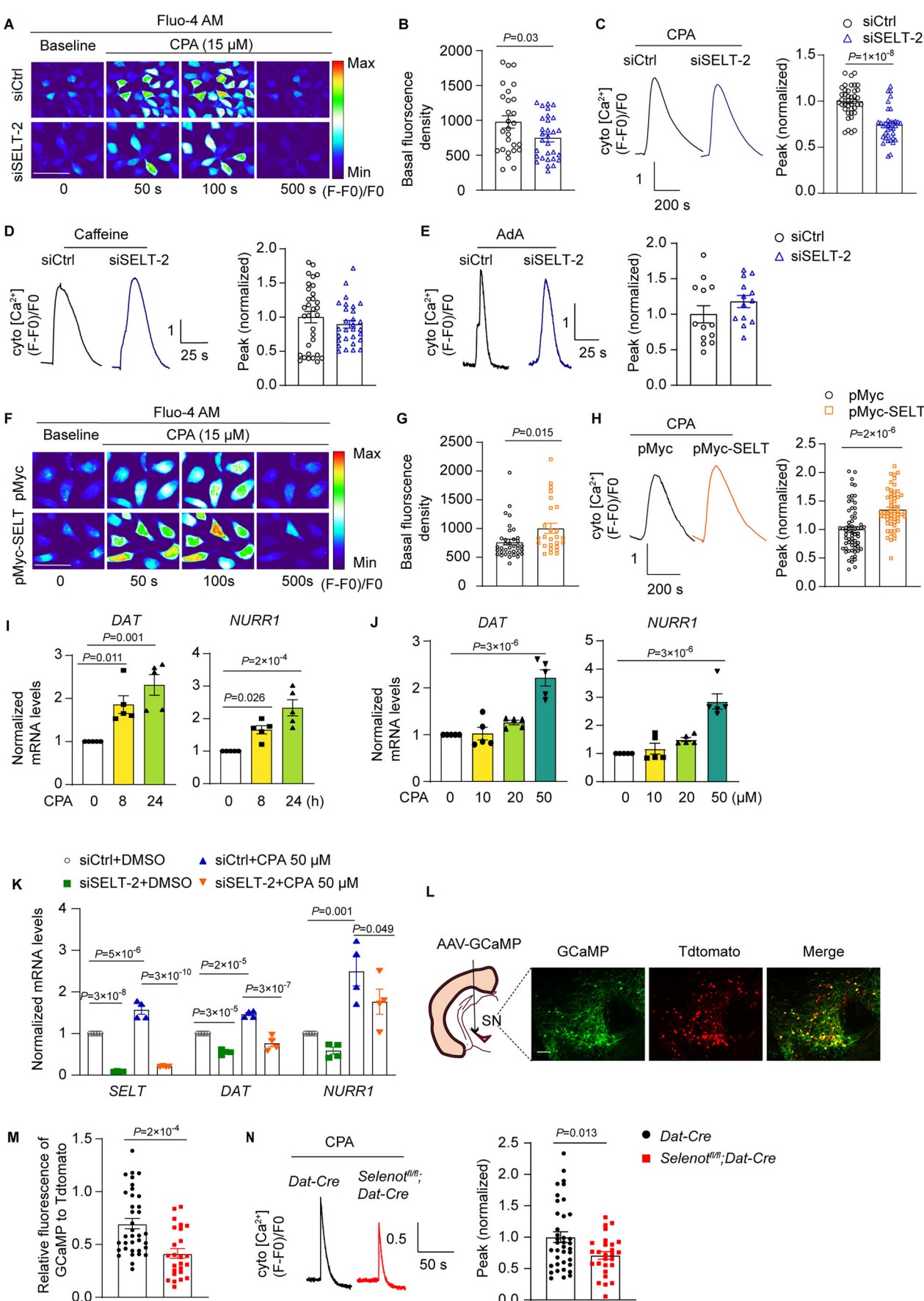

**Figure 7. SELENOT modulates ER-cytosol Ca²⁺ influx via SERCA.**

(A–E) Cytosolic $Ca^{2+}$ level analyses of HEK293 cells transfected with siRNA for 48 h. Presented are representative images (A), steady-state levels (B), CPA-induced peaks (C), caffeine-induced peaks (D), and AdA-induced peaks (E). Data were quantified from 3 independent experiments: $n = 27$ siCtrl and 30 siSELT-2 cells for (B), $n = 30$ siCtrl and 29 siSELT-2 cells for (C), $n = 32$ siCtrl and $n = 29$ siSELT-2 cells for (D), $n = 13$ siCtrl and $n = 13$ siSELT-2 cells for (E). Scale bar, 50 μm. (F–H) Cytosolic $Ca^{2+}$ level analyses of HEK293 cells transfected with plasmid for 48 h. Presented are representative images (F), steady-state levels (G), and CPA-induced peaks (H). Data were quantified from 3 independent experiments: $n = 36$ pMyc and 26 pMyc-SELT cells for (G), $n = 65$ pMyc and 58 pMyc-SELT cells for (H). Scale bar, 50 μm. (I, J) Time-(I) and dose-dependent (J) effects of CPA on *DAT* and *NURR1* expression. HEK293 cells were treated with 50 μM CPA for different time (I), or with different dose of CPA for 24 h (J). Quantifications are normalized to β-actin. $n = 5$ independent experiments. (K) Effect of CPA treatment on *DAT* and *NURR1* expression in *SELENOT* pre-knockdown cells. Quantifications are normalized to β-actin. $n = 4$ independent experiments. (L–N) Cytosolic $Ca^{2+}$ level analyses of substantia nigra slices from mice injected with cre-initiated AAV-GCaMP. Presented are the injection diagram and representative images (L), steady-state levels (M), and CPA-induced peaks (N). Data were quantified from 3 *Dat-cre* and 3 *Selenot^{fl/fl};Dat-cre* mice: $n = 36$ *Dat-cre* and 25 *Selenot^{fl/fl};Dat-cre* cells for (M), $n = 38$ *Dat-cre* and 27 *Selenot^{fl/fl};Dat-cre* cells for (N). Red (Tdtomato) labels nuclei. Quantifications of GCaMP are normalized to Tdtomato. Scale bar, 100 μm. Data are presented as means ± SEM and analyzed by one-way ANOVA followed by Dunnett's post hoc test for (I, J), factorial ANOVA for (K), and two-tailed unpaired *t*-test for other comparisons. AAV adeno-associated virus, AdA adenophostin A, CPA cyclopiazonic acid, pMyc pCMV-Myc empty vector, pMyc-SELT Myc-tagged SELENOT vector, siCtrl scramble siRNA, siSELT *SELENOT* siRNA. Source data are available online for this figure.

---

involved in vivo to maintain the ER homeostasis. Thus, the slight decrease in dopaminergic neuron number in the midbrain may result from the proliferation arrest of progenitor cells during embryonic development (Shao et al, 2019). Nonetheless, this mild decrease does not impact nigrostriatal innervation or dopamine synthesis. Another important implication of the current study is the key role of $Ca^{2+}$ homeostasis in ADHD. This role has not been previously emphasized, except for occasional reports showing genetic associations between this disorder and voltage-gated calcium channels (Andrade et al, 2019; Cross-Disorder Group of the Psychiatric Genomics Consortium, 2019), as well as lower serum $Ca^{2+}$ levels in affected children (Kamal et al, 2014). Additional changes in the dopaminergic system include MAO-A upregulation and enhanced dopamine metabolism. Dopamine is metabolized primarily by MAO-B in humans but by MAO-A in rodents (Cases et al, 1995; Fowler and Benedetti, 1983; Gaweska and Fitzpatrick, 2011). It remains unclear what the increased dopamine metabolism means in this context, but brain MAO-A upregulation has been indicated in a variety of psychiatric illnesses and prodromal states, as indicated by brain imaging and post-mortem evidence (Kolla et al, 2016).

ADHD patients often exhibit abnormal EEG patterns, particularly elevated levels of theta power (Kiiski et al, 2020; Tye et al, 2014), and its heritability is higher in the occipital region, as suggested by a multivariate analysis of EEG rhythms (Zietsch et al, 2007). Indeed, our measurements in the occipital region reveal an elevation in the theta power, but not in other frequencies, in *Selenot^{fl/fl};Dat-cre* mice. The disorder is highly heritable, as evidenced by studies of twins, families, and adoptive siblings (Gallo and Posner, 2016). However, no specific genetic mutation has yet been clinically reported to cause ADHD. Known candidate genes, such as *DAT* and *GIT1*, are suggested by genome-wide association study (GWAS), meta-analyses, large-scale linkage studies, and animal model studies (Gallo and Posner, 2016). This contradiction may be due to the high genetic heterogeneity of the disorder and the involvement of environmental and social factors in its etiology. Thus far, *SELENOT* has not been associated with pathogenic mutations in humans (Santesmasses and Gladyshev, 2021). While no brain expression data are available for patients, GWAS results suggest that two *SELENOT* variants may be potentially associated with ADHD risk, including promoter variant rs2204113 ($P = 0.005$) in childhood (Rajagopal et al, 2022), and intronic variant rs143261451 ($P = 0.01$) (Demontis et al, 2021;

Mattheisen et al, 2022). Our animal data provide evidence of a causative effect and a mechanistic connection to this clinically yet-to-be-ascertained pathogenic relationship. Indeed, *SELENOT* is important for both brain and embryonic development, as indicated here and previously (Boukhzar et al, 2016; Castex et al, 2016).

In conclusion, midbrain SELENOT is indispensable for the maintenance of proper dopamine signaling. SELENOT deficiency in dopaminergic neurons, but not in astrocytes, leads to ADHD-like behaviors in mice. The pathology of this condition is mechanistically associated with a cascade involving the loss of interaction and inhibition of SERCA, reduced cytosolic $Ca^{2+}$ levels, ablation of NURR1 and DAT, reduced dopamine reuptake, increased extrasynaptic dopamine retention, and enhanced striatal neurotransmission. Our findings suggest that SELENOT plays a protective role against ADHD pathogenesis and may serve as a target for genetic screening and therapeutics.

## Methods

### Reagents and tools table

| Reagent/Resource | Reference or Source | Identifier or Catalog Number |
|---|---|---|
| **Experimental models** | | |
| *Selenot^{fl/fl}* line | Cyagen | Generated for this study |
| *Dat/Slc6a3-cre line* | Jackson Laboratory | 006660 |
| *Nestin-cre line* | Cyagen | C001446 |
| *Gfap-cre line* | Cyagen | C001062 |
| *Selenot^{fl/fl};Dat-cre mice* | This study | N/A |
| *Selenot^{fl/fl};Nestin-cre mice* | This study | N/A |
| *Selenot^{fl/fl};Gfap-cre mice* | This study | N/A |
| HEK293 | American Type Culture Collection | CRL-1573 |
| **Recombinant DNA** | | |
| pMyc-SELT | (Shao et al, 2019) | Myc-tagged SELENOT expression vector |

| Reagent/Resource | Reference or Source | Identifier or Catalog Number |
|---|---|---|
| pMyc | (Shao et al, 2019) | pCMV-Myc |
| pMyc-SELT$^{U49C}$ | This study | Myc-tagged SELENOT$^{U49C}$ vector |
| pFlag-SERCA2 | WZ Biosciences | Flag-tagged SERCA2 vector |
| AAV-Ctrl | BrainVTA | rAAV-mTH-EGFP-WPRE-SV40 polyA |
| AAV-SELT | BrainVTA | rAAV-mTH-EGFP-P2A-3×Myc-SELENOT-WPRE-SV40 polyA; constructed for this study |
| AAV-GCaMP | OBiO Technology | pAAV-EF1a-DIO-GCaMP6s-P2A-NLS-dTomato |
| **Antibodies** | | |
| β-actin (rabbit) | Cell Signaling | 4970S |
| BCL-2 (rabbit) | Abmart | T40056 |
| BIP (rabbit) | Proteintech | 11587-1-AP |
| Caspase 3 (rabbit) | Cell Signaling | 9662S |
| Cleaved caspase 3 (rabbit) | Cell Signaling | 9664S |
| CHOP (mouse) | Cell Signaling | 2895T |
| COMT (rabbit) | Abcam | ab126618 |
| DAT (rat) | Merck | MAB369 |
| DAT (rabbit) | Proteintech | 22524-1-AP |
| D2R (rabbit) | Merck | ab5084P |
| MAO-A (mouse) | Santa Cruz Biotech | sc-271123 |
| MAO-B (mouse) | Santa Cruz Biotech | sc-515354 |
| NURR1 (rabbit) | Proteintech | 10975-2-AP |
| SELENOT (rabbit) | Atlas Antibodies | HPA039780 |
| TH (rabbit) | Proteintech | 25859-1-AP |
| TH (mouse) | Immunostar | 22941 |
| TH (mouse) | Santa Cruz Biotech | sc-25269 |
| VMAT2 (rabbit) | Proteintech | 20873-1-AP |
| SERCA2 (mouse) | Thermo Fisher | MA3-919 |
| IP3R1 (rabbit) | Proteintech | 19962-1-AP |
| RYR2 (rabbit) | Proteintech | 19765-1-AP |
| c-MYC (rabbit) | Proteintech | 10828-1-AP |
| c-MYC (mouse) | Proteintech | 67447-1-IG |
| Anti-mouse IgG | Cell Signaling | 7076 |
| Anti-rabbit IgG | Cell Signaling | 7074 |
| Alexa Fluor 555 anti-rabbit IgG | Thermo Fisher | A21429 |
| Alexa Fluor 488 anti-mouse IgG | Thermo Fisher | A11001 |
| **Oligonucleotides and other sequence-based reagents** | | |
| Selenot-genotyping-F | Tsingke | TGCATTTTTCTGTCTCTGGGGAAG |
| Selenot-genotyping-R | Tsingke | CCTCCACGAACAGAGAAGTCAAAGG |
| Cre-genotyping-F | Tsingke | GACCAGGTTCGTTCACTCA |
| Cre-genotyping-R | Tsingke | TAGCGCCGTAAATCAAT |
| Human DAT-qPCR-F | Tsingke | TTCTCCTGTCCGTCATTGGC |
| Human DAT-qPCR-R | Tsingke | CGTGAAGCCCACACCTTTCA |
| Mouse Dat-qPCR-F | Tsingke | TTCATGGTTATTGCCGGGATG |
| Mouse Dat-qPCR-R | Tsingke | TGTAGAAGAAGCCCACGTAGAA |
| Human ACTB-qPCR-F | Tsingke | TGGCACCCAGCACAATGAA |
| Human ACTB-qPCR-R | Tsingke | CTAAGTCATAGTCCGCCTAGAAGCA |
| Mouse Actb-qPCR-F | Tsingke | GCAGGAGTACGATGAGTCCG |
| Mouse Actb-qPCR-R | Tsingke | ACGCAGCTCAGTAACAGTCC |
| Human NURR1-qPCR-F | Tsingke | ACCACTCTTCGGGGAGAATACA |
| Human NURR1-qPCR-R | Tsingke | GGCATTTGGTACAAGCAAGGT |
| Mouse Nurr1-qPCR-F | Tsingke | CAGCTCCGATTTCTTAACTCCAG |
| Mouse Nurr1-qPCR-R | Tsingke | AGGGGCATTTGGTACAAGCAA |
| siCtrl siRNA | GenePharma | UUCUCCGAACGUGUCACGUTT |
| siSELT-1 siRNA | GenePharma | GTCAGTCTTCAAACTAGTATT |
| siSELT-2 siRNA | GenePharma | GCUUCUGCUGCUUCUCCUAUU |
| siNURR1-1 siRNA | GenePharma | AAGAAGTGGTTCGCACAGACA |
| siNURR1-2 siRNA | GenePharma | AGAAAUCGGAGCUGUAUUCUC |
| **Chemicals, Enzymes and other reagents** | | |
| AMPH | Sigma-Aldrich | A-007 |
| Caffeine | Sigma-Aldrich | C0750 |
| Dopamine | Sigma-Aldrich | H8502 |
| CPA | Sigma-Aldrich | C1530 |
| MPH | Rhawn Reagent | 298-59-9 |
| AdA | Santa Cruz Biotech | sc-221213 |
| Biocytin | Thermo Fisher | 28022 |
| BCA kit | Beyotime | P0010 |
| LumiGLO Reagent and Peroxide Kit | Cell Signaling | 7003 |
| Fetal bovine serum | Wisent | 086-150 |
| Lipofectamine 3000 | Invitrogen | L3000015 |
| Protease and phosphatase inhibitors | Beyotime | P0013 |

| Reagent/Resource | Reference or Source | Identifier or Catalog Number |
| --- | --- | --- |
| RNAiso Plus | Takara | 9108 |
| HiScript RT SuperMix | Vazyme | R222-01 |
| Universal SYBR Master Mix | Vazyme | Q711-02 |
| Fluo-4-AM | Beyotime | S1060 |
| **Software** | | |
| Image-pro Plus 6.0 | (Wang et al, 2009) | N/A |
| Image J | Image J | N/A |
| OmicStudio | LC Sciences | https://www.omicstudio.cn/tool |
| SPSS version 23.0 | IBM | N/A |
| **Other** | | |
| ANY-maze video tracking system | Stoelting Co | N/A |
| Rodent brain matrix | RWD Life Science | N/A |
| Eclipse Ti microscope | Nikon | N/A |
| A1R-SIM-STORM microscope | Nikon | N/A |
| CFX Connect System | Bio-Rad Laboratories | N/A |
| Flaming-Brown micropipette puller | Sutter Instruments | P-1000 |
| Vibratome | Leica | VT1000S |
| MultiClamp 700B amplifier and 1550B digitizer | Molecular Device | N/A |
| EEG/EMG system | Pinnacle | N/A |
| HiSeq X10 system | Illumina | N/A |
| Microdialysis probe | Harvard Bioscience | CMA-7 |
| UPLC BEH C18 column | Waters | 186002352 |
| 1290 Infinity II UHPLC and 6470A | Agilent | N/A |
| QTRAP 6500+ system | AB Sciex | N/A |

## Mice

The C57BL/6 line of *Selenot*^fl/fl mice was generated by flanking *Selenot* exons 2 and 3 with *loxP* sites. The *Dat/Slc6a3-cre*, *Nestin-cre*, and *Gfap-cre* lines contain knock-in alleles of Cre recombinase, driven by specific promoters for gene knockout in dopaminergic neurons, in the central and peripheral nervous system including neuronal and glial cell precursors, and in astrocytes, respectively. The Cre lines were first crossbred with *Selenot*^fl/fl mice to generate *Selenot*^fl/fl;*Dat-cre*, *Selenot*^fl/fl;*Nestin-cre*, and *Selenot*^fl/fl;*Gfap-cre* mice. These mouse lines were then re-bred with *Selenot*^fl/fl mice to produce littermates with an identical genetic background. As such,

the Cre genotypes in the conditional knockout lines were all heterozygous. Mice (3–4 per cage) were given free access to pelleted chow diet and distilled water in a specific pathogen-free animal facility maintained at 22 °C with 40–60% humidity and a 12-h light-dark cycle. Unless otherwise indicated, male mice aged 8–12 weeks were used and sacrificed under deep isoflurane anesthesia. The animal experiments were approved by the Institutional Laboratory Animal Care and Use Committee of Wenzhou Medical University (reference No. wydw2019-0678) and conducted in accordance with China National Standard Guidelines for Laboratory Animal Welfare.

## Behavioral assessments

Behavior parameters were recorded and analyzed using the ANY-maze video tracking system. An experimenter was generally blind to the genotypes during the test and animal order was randomly assigned. For the open-field test, spontaneous locomotor activity was recorded for 60 min in a $50 \times 50 \times 50$ cm open-field chamber. For the elevated plus maze, the apparatus was opaque and consisted of a central platform ($10 \times 10$ cm), two open arms ($50 \times 10$ cm), and two closed arms ($50 \times 10$ cm) with protective walls 40 cm high, along with a supporting rod (height, 50 cm). The mouse was placed on an open arm facing another open arm, and activity was recorded for 5 min. For the Y-maze test, the mouse was placed in a Y-shaped maze with three white, opaque arms ($35 \times 6$ cm) positioned at a 120° angle from each other. The mouse was allowed to freely explore the three arms, and activity was recorded for 10 min. For the rotarod test, the mouse was placed on an open rotarod facing away from the experimenter and trained under accelerating condition (4 to 40 rpm over 5 min) for three consecutive days. The latency to fall was recorded on day 4.

For the three-chamber social analysis, the apparatus was a $60 \times 42 \times 22$ cm box divided into three equal-sized compartments with doors between them. The mouse was assessed in the three chambers across three sessions. In session 1, the test mouse was allowed to freely explore the center chamber for 5 min to habituate, with the doors to the side chambers closed. In session 2, a social mouse (stranger 1) in a wire cage was placed into one of the side chambers, and the test mouse was allowed to explore all three chambers for 10 min with the doors open. In session 3, a new mouse (stranger 2) in a wire cage was placed into the other side chamber, and the test mouse was allowed to freely explore for 10 min. The stimulus mice were age- and sex-matched and unknown to the test mouse. The time spent exploring each cage (within a 3-cm range of the cage) was measured. A social preference index was calculated as the ratio of time spent exploring the stranger 1 cage to the total time spent exploring both stranger 1 and the empty cages. A social recognition index was calculated as the ratio of time spent exploring the stranger 2 cage to the total time spent exploring both stranger 1 and stranger 2 cages.

For the novel object recognition, the arena was a $50 \times 50 \times 50$ cm open-field chamber. On day 1, the mouse was allowed to explore freely in the field for 10 min. On day 2, the mouse was introduced to two identical objects in the field and allowed to explore them freely for 10 min. The mouse was then returned to the field after 1 h (short-term) and 24 h (long-term), with one of the objects replaced by a differently shaped object. The time spent exploring each object

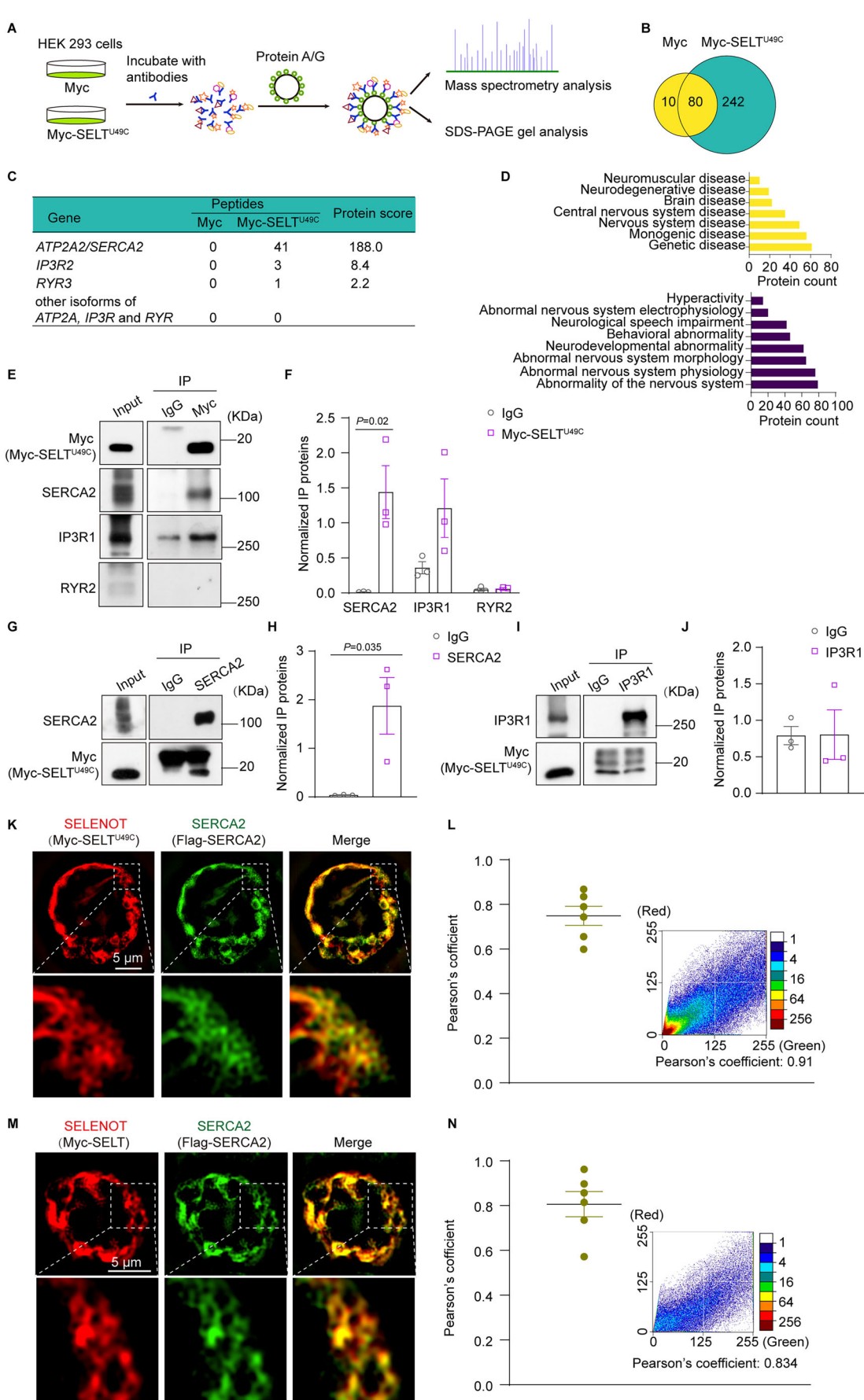

**Figure 8. SELENOT interacts with SERCA2, but not IP3R1 or RYR2.**

(A) Schematic diagram illustrating the identification of SELENOT-interacting proteins. (B) Venn diagram of proteins identified by mass spectrometry of immunoprecipitants from Myc and Myc-SELT[U49C]. Numbers indicate number of proteins pulled down with each bait protein (Protein score > 30). (C) Enriched peptide numbers and protein scores for ATP2A/SERCA, IP3R, and RYR isoforms. (D) Clustering of enriched human phenotypes and diseases among the 242 unique SELENOT-interacting proteins. (E, F) Immunoprecipitation with Myc-SELT[U49C] to detect SERCA2, IP3R1 and RYR2. (G–J) IP with SERCA2 or IP3R1 to detect Myc-SELT[U49C]. HEK293 cells were transfected with pMyc-SELT[U49C] or pMyC for 48 h. Presented are the representative blots (E, G, I) and quantifications normalized to the non-IgG bait (F, H, J). $n = 3$ independent experiments. (K–N) Colocalization analyses of SERCA2 with Myc-SELT[U49C] and Myc-SELT. HEK293 cells were co-transfected with plasmids pFlag-SERCA and pMyc-SELT[U49C] or pMyc-SELT for 48 h. Presented are the representative colocalization images (K, M), and representative Pearson's coefficient analysis with the calculated values (L, N). $n = 6$ cells from 3 independent experiments. Data are presented as means ± SEM and analyzed by two-tailed unpaired $t$-test. flag-SERCA flag-tagged SERCA, IP immunoprecipitation, IP3R inositol 1,4,5-triphosphate receptor, pFlag-SERCA2 Flag-tagged SERCA2 vector, pMyc pCMV-Myc empty vector, pMyc-SELT Myc-tagged SELENOT vector, pMyc-SELT[U49C] Myc-tagged SELENOT[U49C] vector, RYR ryanodine receptor, SERCA sarco-ER Ca[2+] ATPase. Source data are available online for this figure.

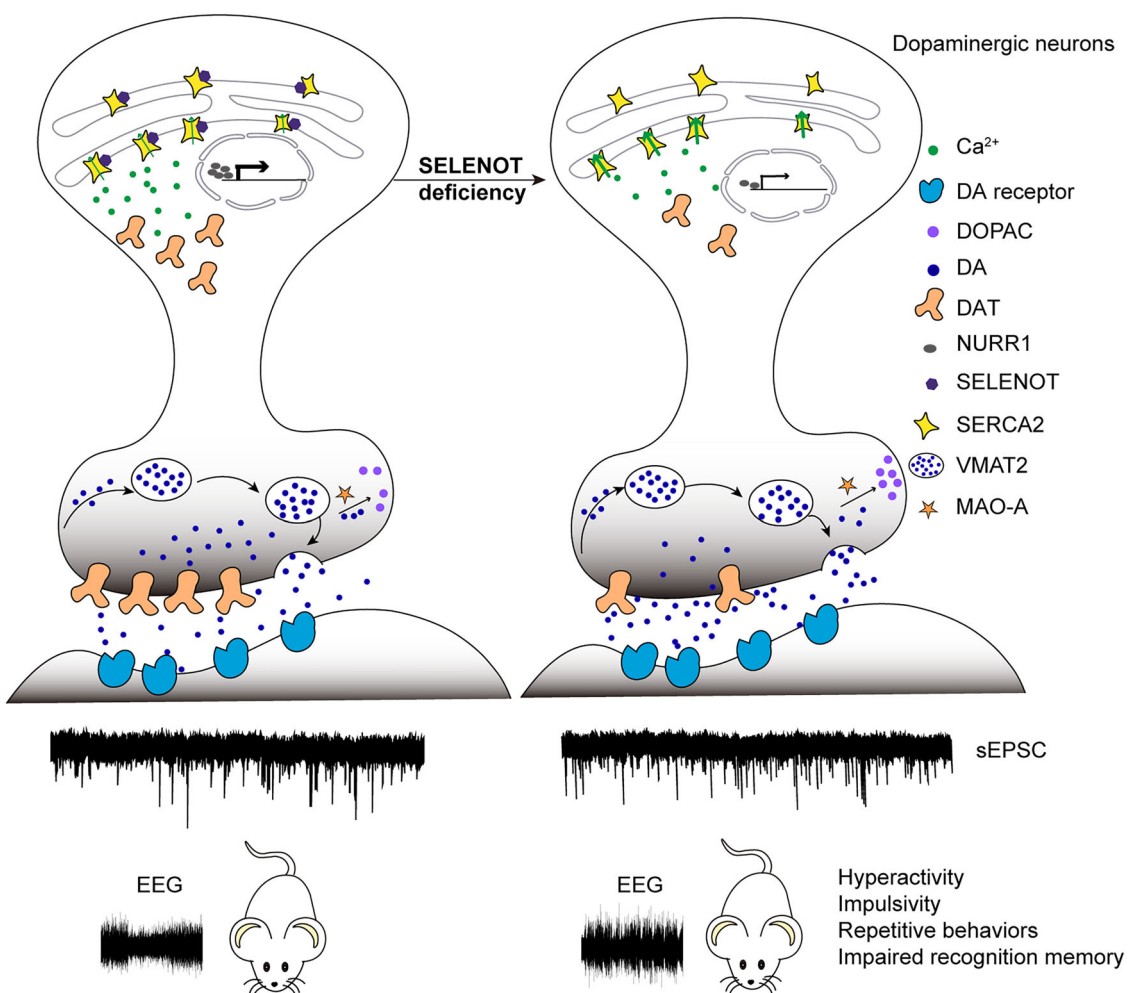

**Figure 9. Working model for SELENOT deficiency inducing ADHD-like behaviors.**

DAT dopamine transporter, DOPAC 3,4-dihydroxyphenylacetic acid, DA dopamine, EEG electroencephalogram, MAO-A monoamine oxidase, NURR1 nuclear receptor-related 1, SELENOT selenoprotein T, SERCA2 sarco-ER Ca[2+] ATPase 2, sEPSC spontaneous excitatory postsynaptic currents, VMAT2 vesicular monoamine transporter 2.

was measured. Object exploration was defined as the mouse being within a 3-cm range to the object. Mice with total exploration times less than 5 sec were excluded from the analysis. Novel object preference was defined as the percentage of time spent exploring the novel object relative to the total object exploring time. For the repetitive behavior test, the mouse was placed in a plastic cage containing 15 cm of corn cob bedding with a lid and recorded for

its behavior for 20 min. The incidences of rearing, digging, tail flicking, climbing, and jumping were counted. For the cliff avoidance reaction, the apparatus consisted of a round plastic platform (diameter, 20 cm; thickness, 2 cm), stably supported by a plastic rod (height, 50 cm). The mouse was placed on the edge of the platform. Impulsivity was expressed as the frequency of the mouse falling from the platform, recorded within 60 min.

## Tissue collection

Mice were euthanized and transcardially perfused with 10 mL of saline. The brain was dissected, rinsed with cold water, and immediately placed in a chilled rodent brain matrix. The brain was then sliced with a cold razor blade, starting ~1 mm rostral to the lambda and sectioning into 1-mm thickness for the substantia nigra and striatum. The desired tissue was collected from the slices and rapidly frozen in liquid nitrogen before being stored at −80 °C for further use.

## Western blot

Cells or tissues were lysed in a sample buffer (60 mM Tris-HCl, pH 6.8; 2% SDS; 5% glycerol) and boiled for 10 min. After centrifugation, the total protein concentration was measured using a BCA kit. An equal amount of total protein from each sample was loaded and analyzed by Western blot as previously described (Zhu et al, 2011). Membranes were cut to probe multiple proteins based on molecular weight for efficiency, as well as to conserve samples and antibodies. The β-actin cropped from the same gel or from a gel run parallelly in the same setting was used as the control in general.

## Immunohistochemistry and immunofluorescence

Mice were euthanized and transcardially perfused sequentially with saline (10 mL) and 4% paraformaldehyde (5 mL). The brains were removed and fixed in 4% paraformaldehyde for 24 h before being dehydrated in ethanol, embedded in paraffin, and sectioned to 5 μm. Endogenous peroxidase activity was blocked with 3% $H_2O_2$ at room temperature for 20 min. The brain slices were blocked in 5% bovine serum albumin for 30 min and incubated with primary antibodies at 4 °C overnight. For immunohistochemistry, the slices were incubated with horseradish peroxidase-conjugated secondary antibodies for 1 h at room temperature. 3,3'-diaminobenzidine was applied to visualize immunohistochemical staining. Nuclei were counterstained with hematoxylin before being mounted with neutral gum. For immunofluorescence, the slices were incubated with secondary antibodies for 2 h at room temperature and counterstained with Hoechst 33258 for 5 min to locate the nuclei before being mounted with anti-fade mounting media. Images were visualized using the Eclipse Ti or A1R-SIM-STORM microscope and quantified using Image-pro Plus 6.0 for immunohistochemistry (Wang et al, 2009) and Image J for immunofluorescence.

## Transfection

HEK293 cells were cultured in DMEM supplemented with 10% fetal bovine serum and maintained in a humidified incubator at 37 °C with 5% $CO_2$. Cells were transfected with siRNA or plasmids for the desired duration, as detailed in the figure legends, using Lipofectamine 3000. The Myc-tagged SELENOT expression vector (pMyc-SELT) was constructed using the pCMV-Myc plasmid (pMyc), as previously described (Shao et al, 2019). The Myc-tagged SELENOT$^{U49C}$ vector (pMyc-SELT$^{U49C}$) was constructed using the overlap extension PCR method.

## Co-immunoprecipitation

After transfection with pMyc or pMyc-SELT$^{U49C}$ for 48 h, HEK293 cells were lysed in lysis buffer supplemented with protease and phosphatase inhibitors. Cell lysates were incubated with IgG or antibodies against Myc, SERCA2, or IP3R1 at 4 °C overnight. An equal volume of protein A/G agarose beads was then added to the lysates and incubated at 4 °C for 2 h. After centrifugation, the precipitated beads were washed with PBS for 5 times. The bound proteins were eluted with 1× loading buffer and sent to BGI Genomics (Shenzhen, China) for identification via liquid chromograph liquid chromatography-tandem mass spectrometry (LC-MS/MS). The expression and immunoprecipitation of Myc-SELENOT were confirmed by both silver staining and Western blot in pre-experiments.

## Real-time PCR

RNA was isolated using RNAiso Plus and reverse transcribed to cDNA using the HiScript RT SuperMix. Quantitative mRNA levels were assayed in triplicate using the Universal SYBR Master Mix following the manufacturer's instructions on the CFX Connect System. Expression levels were calculated as $2^{-\Delta\Delta Ct}$, with *ACTB* as an internal control.

## Electrophysiology

Transverse brain slices (300 μm thick) were freshly prepared in an ice-cold dissection buffer (211 mM sucrose, 3.3 mM KCl, 1.3 mM $NaH_2PO_4$, 0.5 mM $CaCl_2$, 10 mM $MgCl_2$, 26 mM $NaHCO_3$, and 11 mM glucose) using a Vibratome. Slices were then recovered in a holding chamber containing oxygenated (95% $O_2$, 5% $CO_2$) artificial cerebrospinal fluid (oxy-aCSF; 124 mM NaCl, 3.3 mM KCl, 2.5 mM $CaCl_2$, 1.5 mM $MgCl_2$, 1.3 mM $NaH_2PO_4$, 26 mM $NaHCO_3$, and 11 mM glucose) for 1 h at 33 °C. Borosilicate glass pipettes with resistances in the 5–8 MΩ range were pulled using a Flaming-Brown micropipette puller. Using the pipettes, dopaminergic neurons in the substantia nigra were recorded based on morphology and location, and striatal neurons in the dorsal striatum were also recorded, with most of these neurons being medium spiny neurons by morphology. However, we did not further distinguish them as D1R-positive or D2R-positive neurons. Data were acquired and analyzed using pClamp10 and Clampfit 10 on a MultiClamp 700B amplifier and a 1550 B digitizer, respectively.

## Virus-mediated SELENOT in vivo expression

Expression of the adeno-associated viruses (AAVs), rAAV-mTH-EGFP-WPRE-SV40 polyA (AAV-Ctrl) and rAAV-mTH-EGFP-P2A-3×Myc-SELENOT-WPRE-SV40 polyA (AAV-SELT), was initiated under the mouse TH promoter. Herein, the selection of TH promoter over DAT was based on the company's instruction indicating that the AAV with TH promoter has been technically more mature. Mice were stereotaxically injected with 2 μL of AAV ($1.3 \times 10^{12}$ vector genomes/mL) over 10 min into the substantia nigra [from bregma: anterior-posterior (AP) −3.0 mm, media lateral (ML) ± 1.2 mm, and dorsal-ventral (DV) −4.5 mm], after which the needle was left in place for an additional 5 min. Mice underwent behavior test 2 weeks post-administration and were sacrificed after the test.

## Calcium imaging

The AAV, pAAV-EF1a-DIO-GCaMP6s-P2A-NLS-dTomato (AAV-GCaMP), was used to initiate expression under the control of Cre

recombinase. Thus, *Dat-cre* mice, rather than *Selenot*^fl/fl^ mice, were used as the control. Mice were stereotaxically injected with 2 μL of AAV ($1.5 \times 10^{12}$ vector genomes/mL) into the substantia nigra as described above and sacrificed 3 weeks later. Transverse brain slices were prepared as detailed above in the electrophysiology section. Slices containing the substantia nigra were recovered in a holding chamber containing oxy-aCSF without $CaCl_2$ for 8 min at 33 °C. Time-lapse images were then recorded at 15 frames per second using the A1R/SIM/STORM microscope as follows: recording 1 min at the basal level, adding the drug, incubating for 7 min, and then recording 3 min. A 488 nm and a 561 nm laser were used to excite GCaMP6s and dTomato, respectively. The region of interest (ROI) was defined before quantifying features. Using Time Series Analyzer in Image J software, ROIs were drawn around individual cells, and the fluorescence intensities of the ROIs in the recorded video images were automatically quantified. Data were then acquired and statistically analyzed.

For cellular calcium imaging, cells were washed and incubated with 2 μM Fluo-4-AM in HBSS (137 mM NaCl, 0.3 mM $Na_2H$-$PO_4 \cdot 12H_2O$, 5.4 mM KCl, 0.4 mM $KH_2PO_4$, 5.6 mM D-Glucose, 4.2 mM $NaHCO_3$) for 30 min at 37 °C. After washing with HBSS 3 times, cells were then subjected to time-lapse imaging by recording 1 min at the basal level, followed by 5 min after drug administration, and analyzed as described above.

## EEG recording

Mice were anesthetized with 1% (w/v) sodium pentobarbital. Epidural electrodes were bilaterally fixed to the skull with screws and dental cement. Electromyogram electrodes were placed in the neck muscles. After a 1-week recovery period, EEG recording was performed on the freely moving mouse at 500 Hz using the EEG/EMG system. Recorded traces were high-pass filtered at 3 Hz, and the first hour of recording was analyzed for power spectral density.

## RNA sequencing

Total RNA was extracted using RNAiso Plus following the manufacturer's protocol. RNA sequencing was performed using the HiSeq X10 system at LC Sciences (Hangzhou, China), as previously described (Li et al, 2022). After removing low-quality reads, HISAT package was used to map the reads to the *Mus musculus* reference genome (http://genome.ucsc.edu/). The mapped reads were assembled using StringTie and merged to reconstruct a comprehensive transcriptome with Perl scripts. StringTie was then used to determine mRNA expression levels by calculating fragments per kilobase per million mapped reads. Differentially expressed mRNAs were selected by DESeq2 with a fold change >1.2 and $P < 0.05$. Bioinformatic analyses were performed using the OmicStudio tools with significance set at $P < 0.05$.

## In vivo microdialysis

The mouse was stereotaxically implanted with a microdialysis probe in the striatum (from bregma: AP + 0.6 mm, ML −1.8 mm, and DV −4.0 mm) under 1% (w/v) sodium pentobarbital anesthesia. After a 12-h recovery, the probe was continuously perfused with artificial cerebrospinal fluid (147 mM NaCl, 3.5 mM KCl, 1.2 mM $CaCl_2$, 1.2 mM $MgCl_2$, 1 mM $NaH_2PO_4$, and 25 mM $NaHCO_3$, pH 7.0–7.4) at a flow rate of 1 μL/min. Ten samples were collected at 35-min intervals. The first 4 were discarded as equilibration samples, and the remaining 6 samples were stored at −80 °C for dopamine analysis. The location of the probe within the striatum was confirmed histologically.

## Neurotransmitter measurement

Neurotransmitters and metabolites were analyzed by ultra-high-performance LC-MS/MS with quality control assessment. All internal standards were obtained from Sigma. Samples were separated using a UPLC BEH C18 column ($100 \times 2.1$ mm, 1.7 μm). Striatal sample preparation and measurement were performed at ProfLeader Biosciences (Shanghai, China) using a coupled system of 1290 Infinity II UHPLC and 6470 A mass spectrometer. Microdialysis samples for dopamine analysis were measured using the QTRAP 6500+ system.

## Statistical analysis

Sample size estimations were based on previous studies (Li et al, 2017; Pristera et al, 2019). Statistical differences were evaluated using Student's *t*-test, one-way analysis of variance (ANOVA) followed by Dunnett's post hoc test, repeated measures ANOVA or factorial ANOVA followed by Fisher's LSD post hoc test, as specified in the figure legends. Data are presented as mean ± standard error of the mean (SEM). Differences were considered statistically significant at $P < 0.05$. All analyses were conducted using SPSS version 23.0.

# Data availability

The bulk RNA-seq data are deposited in the Sequence Read Archive database under the accession number PRJNA910327.

The source data of this paper are collected in the following database record: biostudies:S-SCDT-10_1038-S44318-025-00430-3.

# Peer review information

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

## Acknowledgements

The authors are grateful to Dr. Jiang-Fan Chen for assistance in reagents and the Scientific Research Center of Wenzhou Medical University for consultation and instrument support. This work was partially supported by National Natural Science Foundation of China (82071585, 82271282, and 82471617) and Zhejiang Provincial Natural Science Foundation (ZCLTGD24C0902, ZCLZ25H2502, and LZ25H090005).

## Author contributions

**Qing Guo**: Data curation; Formal analysis; Validation; Investigation; Visualization; Methodology; Writing—original draft; Project administration. **Zhao-Feng Li**: Formal analysis; Validation; Investigation; Methodology. **Dong-Yan Hu**: Data curation; Formal analysis; Validation; Investigation; Visualization. **Pei-Jun Li**: Formal analysis; Investigation. **Kai-Nian Wu**: Data curation; Formal analysis; Validation; Investigation. **Hui-Hui Fan**: Resources; Funding acquisition; Project administration. **Jie Deng**: Resources; Writing—review and editing. **Hong-Mei Wu**: Resources; Formal analysis. **Xiong Zhang**: Conceptualization; Supervision; Funding acquisition; Writing—review and editing. **Jian-Hong Zhu**: Conceptualization; Resources; Supervision; Funding acquisition; Writing—review and editing.

Source data underlying figure panels in this paper may have individual authorship assigned. Where available, figure panel/source data authorship is listed in the following database record: biostudies:S-SCDT-10_1038-S44318-025-00430-3.

## Disclosure and competing interests statement

The authors declare no competing interests.

