## [Peer Review File · The EMBO Journal]

The selenocysteine-containing protein SELENOT maintains dopamine signaling in the midbrain to protect mice from hyperactivity disorder

Qing Guo, Zhao-Feng Li, Dong-Yan Hu, Peijun Li, Kai-Nian Wu, Hui-Hui Fan, Jie Deng, Hong-Mei Wu, Xiong Zhang, and Jian-Hong Zhu

Corresponding author(s): Jian-Hong Zhu (jhzhu@wmu.edu.cn) , Xiong Zhang (xiongzhang@wmu.edu.cn)

Review Timeline:

Submission Date:	18th Jan 24
Editorial Decision:	15th Apr 24
Appeal Received:	6th Feb 25
Editorial Decision:	4th Mar 25
Revision Received:	10th Mar 25
Accepted:	14th Mar 25

Editors: Kelly M Anderson and Ioannis Papaioannou

Transaction Report:

Dear Dr. Zhu,

Thank you for submitting your manuscript for consideration by the EMBO Journal. It has now been seen by two referees whose comments are shown below.

Given that the referees both recommend a large number of experiments that would be required to satisfy their concerns, we unfortunately do not feel this would be feasible to accomplish within our normal revision time of 3-5 months and therefore, I am afraid we cannot offer to publish it here. That said given the interest in this area, we would be willing to consider a revised version at a later date if you are able to address all of the concerns, however we would re-assess the novelty and interest in the story again at the time of re-submission.

Thank you in any case for the opportunity to consider this manuscript. I am sorry we cannot be more positive on this occasion, but we hope nevertheless that you will find our referees' comments helpful.

Yours sincerely,

Kelly M Anderson, PhD
Editor, The EMBO Journal
k.anderson@embojournal.org

Referee #2:

This manuscript from Guo et al asks how the loss of ER protein Selenoprotein T (Selenot) dysregulates dopaminergic function and "ADHD-like" behaviors in male mice. Central Selenot knockout, or Selenot loss in dopamine neurons caused a reduction in dopamine transporter expression (DAT), and a hyperlocomotion and increased stereotypy behavioral phenotype that is typical of DAT knockout mice. The hyperlocomotion phenotype is blocked with higher doses of psychostimulants amphetamine and methylphenidate. In mice lacking Selenot in DAT⁺ cells, there is increased striatal dopamine metabolites and basal dopamine tone, as well as increased excitatory input onto striatal medium spiny neurons, and reduced expression of midbrain Nr4a2. To look at the molecular mechanisms, the authors use siRNA against Selenot, or Selenot overexpression in HEK293 cells, finding DAT and Nr4a2 expression are linked to the degree of Selenot expression. They find in vitro, that Selenot control over DAT expression is mediated through Nr4a2, as NR4a2 knockdown occluded the effects of Selenot overexpression on DAT mRNA. Using in vitro and ex vivo calcium imaging, they found Selenot expression was linked to cytosolic calcium regulation by SERCA, such that SERCA-inhibition-induced cytosolic calcium was potentiated in Selenot overexpressing HEK293 cells and attenuated in the Selenot knockdown or floxed mouse midbrain. In a series of immunoprecipitation experiments, they confirm Selenot has stronger interaction with SERCA vs IP3 or ryanodine receptors.

This manuscript provides important insight into regulation of dopamine transporter expression, which has implications for a wide variety of neurodegenerative and neuropsychiatric disorders. My primary concerns can be summarized as (1) The organization of the manuscript and figures makes the story difficult to follow (2) The over-reliance on representative images without quantification/insufficient validation (3) No Selenot overexpression in vivo to corroborate that Selenot directly controls DAT expression via Nr4a2, experiments that are necessitated by the selection of HEK293 cells, which typically exhibit very low levels of DAT expression.

1. The manuscript is extremely difficult to follow as the authors switch back and forth between in vivo and in vitro experiments in the same figure.
 - (a) I recommend moving Figure 5 (Selenot-flox-Nestin-Cre) to the supplement. Then, the idea that the behavioral effects are neuron specific ("Selenot deficiency in whole brain, ...") can follow the first section in the results that describes the behavioral effects in the Selenot-Flox-DAT-Cre (Selenot^{fl/fl};Dat-cre mice display such ADHD-like...).
 - (b) Figure 9, with the behavioral rescue with amph/mph, should be additional panels in Figure 1, with perhaps the lower doses (9ABEF) in the supplement.
 - (c) New Figure 5 should break out the mouse RNAseq into its own figure, showing the volcano (6D), the GO analysis (7A), the heatmaps (6E), the ADHD-associated gene violin plots (6F), and the protein-level validation of DAT and Nr4a2 (6GHI)
 - (d) New Figure 6 becomes the Selenot knockdown and overexpression, and Nr4a2 knockdown occlusion experiment, all in HEK293
 - (e) New Figure 7 would better be organized as the HEK293 experiments (knockdown, overexpression, CPA-dependent transcription), then the slice experiments.

(f) New Figure 8 would remove panel D with the RNAseq counts from the mice, leaving the figure all HEK293.

2. The following quantifications or validations are missing or inadequate:

(a) Selenot mRNA for siSELT1, siSELT2 in HEK293

(b) Selenot protein for siSELT1, siSELT2 in HEK293- there's only representative blot in 6A

(c) Quantification of the IP experiments in Fig 8FG-there's only representative blots

(d) Quantification of the colocalization in Fig 8HI-there's only representative IHC

(e) Selenot mRNA/protein in the Selenot-flox-DAT-cre animals. There are RNAseq counts in the supplement, but it is unclear if they are different, and they may not be due to bulk RNAseq. An acceptable means of showing decreased Selenot expression in the Selenot-flox-DAT-cre animals would be a quantification of Selenot positive cells as a percentage of TH positive cells-- Figure 1B with TH/Selenot colocalization.

(f) unquantified westerns for Selenot-flox-GFAP-Cre and Selenot-flox-Nestin-Cre. Also, region for the western in Supp 5B is unspecified.

(g) Figure 4K-L. The more appropriate quantification would be number of TH and/or DAT positive nuclei as a percentage of the total nuclei. Fluorescence intensity in IHC is an unreliable indicator of degree of expression.

(i) Figure 7J-K. Fluorescence intensity can change as a function of sensor expression. There is no indication that GCaMP is similarly expressed in the Selenot-flox-DAT-cre vs DAT-cre, rendering the "basal fluorescence density" in 7K uninterpretable. It's also worth mentioning the methodologic details regarding the calcium imaging is lacking-e.g. there are no excitation/emission filters, the analysis is insufficiently described ("with ImageJ")

3. The idea that Selenot directly controls DAT expression via Nr4a2 and SERCA is inadequately supported.

(a) HEK293 cells do not normally express high levels of DAT, and are a poor model for neurons as they are normally electrically unexcitable cells. While many of the siRNA knockdown in HEK293 cells recapitulate the knockout in DAT cells (gene and protein expression, reduced CPA-induced Ca²⁺ influx), there are no experiments showing the inverse-that increasing Selenot in vivo also increases Nr4a2 and DAT expression. This is important since the mice are lacking Selenot in DAT cells from development, and the decrease in DAT and Nr4a2 could arise from some other compensatory mechanism. Viral-mediated Selenot overexpression in vivo should increase Nr4a2 and DAT expression as it does in the in vitro experiments.

(b) Figure 7M-N show that SERCA influences DAT and Nr4a2 expression, not that this is mediated by Selenot. To make that claim, the authors must repeat the CPA-stimulated DAT and Nr4a2 expression in the Selenot knockdown and/or overexpressing cells.

Other concerns

1. Figure 3-the striatal medium spiny neurons have increased excitatory input (greater sEPSC frequency), not increased excitability-to claim increased excitability the authors need to measure rheobase and/or current evoked spikes.

2. Figure 4-- Drd antibodies are notoriously non-specific (see PMID: 2831602). While the Drd2 antibody from Abcam used in this manuscript has been validated with knockout animals, there is no knockout validation for the Drd1 antibody from Santa Cruz. Previous versions of Santa Cruz Drd antibodies continue to show signal in western blot in knockout animals. If the authors cannot find a citation showing this antibody is specific, the band for Drd1 should be removed from the manuscript so as not to perpetuate the flawed rationale that an antibody is ok to use since it's been used in another manuscript.

3. The manuscript seems to have been conducted exclusively in male mice. This should be specified in the abstract.

4. How are "social recognition index" and "social preference index" calculated?

5. Lines 107-112. "suggesting accelerated dopamine metabolism..." ... "indicate that dopamine metabolism was impaired". These two statements contradict one another. Based on the increased dopamine metabolites and MAO-A expression, dopamine metabolism appears to be increased in the selenot-flox-DAT-cre mice. The microdialysis seems to indicate a greater basal dopamine tone in the selenot-flox-DAT-cre mice, which is not explained by increased midbrain dopamine neuron firing, but does make sense given the loss of DAT.

6. There are instances in which unpaired t-tests are used for paired measurements- Supplemental 2J, 2N, 5L,5O

Minor concerns

1. In the abstract, specify that it's the number of dopamine neurons in the SNc that are reduced.

2. Figure 1L heatmap for the selenot-flox-dat-cre, mice appear to spend less time interacting with objects in general. The exploration time in both genotypes should be reported in the supplement, not just the ratio of novel to familiar.

3. Figure 4-what is the arrow on the VStr VMAT2 western?

4. The violin plots of "adhd candidate genes" - it's not really clear from the plots that they are significant from each other and to what extent these represent a large percentage of the candidate genes-i.e. how many ADHD candidate genes are there per DisGenet?

5. Many of the RNAseq visualizations would be more informative if there was a supplemental table of DEGs showing logFC and p-value. The volcano plot is fine, but only two labeled genes, which are then repeated in the adjacent heatmap, isn't very informative.

Referee #3:

General Summary: In their manuscript, Guo et al. presented an extensive study on the role of selenoprotein T (Selenot) on dopamine neuron function, the regulation of DAT through the transcription factor NURR1 (a mechanism involving Ca²⁺ influx to the ER through the SERCA2 pump) and a putative role in attention deficit and hyperactivity disorder (ADHD)-like behaviors. The authors used a Selenot^{fl/fl}; Dat-Cre mouse model for most of their experiments and performed behavioral, electrophysiological (patch-clamp and EEG), HPLC with mass spectrometry, immunohistochemistry and western blot experiments to determine SELENOT role in striatal and midbrain dopamine function. They further tested whether their observations on dopamine function were due to an effect of Selenot deletion on neuronal or astrocytes populations comparing a Selenot^{fl/fl}; Nestin-cre and Selenot^{fl/fl}; Gfap-cre mice strains, respectively. Furthermore, Guo et al., performed RNA seq and qPCR experiments in substantia nigra of Selenot^{fl/fl}; Dat-Cre mice and identified Nurr1 to be downregulated along with Dat. Then, to determine a possible mechanism of SELENOT on DAT expression, they used transfection in HEK293 cultures, calcium imaging and co-immunoprecipitation experiments and determine that SELENOT interact with SERCA2 channels to modulate cytoplasmic Ca²⁺ and the expression of NURR1 and DAT. Finally, the authors rescued the hyperactive phenotype observed in Selenot^{fl/fl}; Dat-Cre mice using amphetamine or methylphenidate treatment. Guo et al., concluded that Selenot in dopaminergic neurons maintains proper dopamine signaling and protects against ADHD-like behaviors.

Opinion: Guo et al., did a great deal of work in this manuscript. However, the link between the putative ADHD-like behaviors on their mouse model and the findings on the interaction of SELENOT with SERCA and Ca²⁺, documented in HEK293 cell cultures, is directly supported only by the data showed in Figure 7J-K.

Primary concern: Although this is an extensive study involving several methodologies and techniques, it does not make a cohesive argument to support that SELENOT is indispensable in protecting against dopamine dysfunction and attention deficit and hyperactivity disorder (ADHD)-like behaviors. First, the authors have similar behavioral observations using Selenot^{fl/fl}; Dat-Cre mice and the less dopamine neuronal-specific Selenot^{fl/fl}; Nestin-Cre mice. There is also the possibility that the hyperactivity and DAT expression anomalies reported in the Selenot^{fl/fl}; Dat-Cre mice are related to the Dat-Cre genotype, even as heterozygous. Please see the comment in the Materials and Methods section below and Machado-Acosta et al. 2021, doi: 10.1038/s41598-021-82600-x).

Minor concern: The manuscript is in general well-written; however, some parts will require proper grammar revision.

Other specific comments and questions follow below:

Title

I am unsure if the title conveys what the authors demonstrated with their Selenot^{fl/fl}; Dat-Cre mouse as a model of ADHD-like behaviors. The author provided evidence to support the interaction of SELENOT with SERCA2 and this could in turn modulate NURR1 and DAT expression, but there is no direct evidence that this mechanism is responsible for ADHD-like behaviors observed in Selenot^{fl/fl}; Dat-Cre mouse.

Abstract

Lines 28-29. The study seems to have dropped the finding of increased monoamine oxidase A in relation to ADHD-like behavior in Selenot^{fl/fl}; Dat-Cre mice. They discussed this in lines 362-367, but it could be removed from the abstract since this was not further studied in the present manuscript.

Introduction

Lines 68-70. This statement is an overstatement since the authors did not provide direct evidence that Selenot deficit only in dopaminergic neurons produced increased hyperactivity. They provided supporting evidence that SELENOT mediated Ca²⁺ homeostasis in HEK293 cells. Selenot^{fl/fl}; Nestin-Cre mice also developed similar hyperactivity behaviors.

Results

Lines 78-79. What is the percentage of colocalization between SELENOT and TH? How substantia nigra (SN) expression of SELENOT is compared to dopamine neurons in the ventral tegmental area (VTA)?

Lines 85-86. The authors report no differences in central open-field crossings, but the activity heatmap examples in Figure 1D, could suggest an increase in crossings in Selenot^{fl/fl}; Dat-cre mice compared with Selenot^{fl/fl} mice. Maybe a better example that reflects the group average is needed in this figure?

Lines 88-89. Could the increase in open arm entries indicate increased impulsivity or just reflect the hyperactivity in Selenot^{fl/fl}; Nestin-Cre mice?

Lines 90-92. The novel object recognition test results can be interpreted as a decrease in object exploration (object avoidance) when looking at the provided heat maps at the 24 hours test in Figure 1N. Selenot^{fl/fl}; Nestin-Cre mice seem to spend more time away from the objects and in the corners of the test chamber.

Lines 92-93 and 96-97. In which context and how repetitive behaviors reported here and in Figure 1P were measured?

Lines 97-98. The heat map provided in Supplementary Figure 2I could suggest that Selenot^{fl/fl}; Dat-Cre mice have higher social interaction time than Selenot^{fl/fl} mice.

Lines 105-111. How do the authors explain increased striatal DA in the micro-dialysis experiments but not in DA tissue content? Do Selenot^{fl/fl}; Dat-Cre mice have an accelerated or impaired dopamine metabolism? The author uses both statements in

different places in the paragraph.

Lines 111-113. The microdialysis technique does not have enough resolution to support the statement of a "high level of dopamine level retained in the synaptic gap in the striatum of Selenotf/f; Dat-Cre mice."

Line 113. Could the increase in glutamic acid indicate tissue damage (i.e. glutamate exocytosis)?

Lines 121-123. How were recordings on sEPSC in medium spiny neurons performed? What was the location and type of medium spiny neuron recorder? There is no description of this in the methods section.

Lines 123-124. Are the authors suggesting increased striatal extrasynaptic dopamine directly increased sEPSC in medium spiny neurons?

Lines 134-145. Do the authors observe any regional differences in markers of DAT or TH expression between the substantia nigra vs. the ventral tegmental area or dorsal vs. ventral striatum within genotypes?

Lines 141-145. Quantification by Western blots is usually considered a better parameter than fluorescence intensity from immunolabeling, particularly to assess changes in protein expression. One reason is unspecific fluorescence labeling, like the one visible in Figure 4K for DAT, which does not seem to correspond to TH-positive cells.

Lines 145-147. Which region of the striatum was used for DAT-Cre vs. Selenotf/f comparison? The blots provided in Figure 4A show that the ventral striatum has higher DAT expression than the dorsal striatum in both Selenotf/f and Selenotf/f; DAT-Cre mice. Also, -actin expression in Supplementary Figure 4C seems lower than in blots provided in other figures.

Lines 161-190. Results showed in Figure 5 related to the behavioral testing, DAT and TH protein expression quantification in substantia nigra and striatum from Selenotf/f; Nestin-Cre mice suggest that the dopamine deficits observed in Selenotf/f; Dat-Cre may not be specific to dopamine circuitry. The authors showed that a similar effect can be achieved with an unspecific neuronal deficit of Selenot.

Lines 180-182. In the social preference and social recognition tests, the position of the stranger mice shown in the heat maps of Supplementary Figures 5K and 5N is different from that in the experiments for Selenotf/f; Dat-Cre (Supplementary Figures 2I and 2M).

Lines 203-207. These results could predict that overexpression of SELENOT can rescue the hyperactive phenotype observed in Selenotf/f; Dat-Cre and Selenotf/f; Nestin-Cre mice lines.

Line 207. Why the authors selected substantia nigra for the RNA-seq experiments? In Selenotf/f; Dat-Cre, DAT was more drastically reduced in striatal tissue. Striatal RNAseq analysis could also provide supporting evidence to corroborate the changes in dopamine metabolism reported in Figure 2F.

Lines 249 and 251. Could CPA treatment rescue the phenotype of Selenotf/f; Dat-Cre mice?

Lines 261-264. Were Serca1/Atp2a1, Ip3r1 and Ryr2 higher in the substantia nigra of Selenotf/f; Dat-Cre mice compared with Selenotf/f mice? Otherwise, this could suggest that the Selenotf/f genetic modification alone modify the expression of these genes and explain the behavioral observations on Selenotf/f; Nestin-Cre mice.

Materials and Methods

Lines 397-403. I cannot find whether the Dat-Cre, Nestin-Cre, or Gfap-Cre genotype was bred to generate homozygous or heterozygous mice. Can the authors provide more details on the source of the Cre lines? It will be essential to know how the authors developed a DAT-Cre line without significantly altering DAT expression. This is important since Dat-Cre mice lines develop hyperactivity and attenuated hyperlocomotion in response to amphetamine and lower DAT expression, even as heterozygous (See, for example, Machado-Acosta et al. 2021, doi: 10.1038/s41598-021-82600-x).

Lines 543-547. What was the read depth on the RNAseq? I read the reference provided but cannot find the read depth there either.

Line 521. I understand the rationale to use DAT-Cre mice as a control in this experiment but also highlight the importance of comparing DAT-Cre mice with the Selenotf/f; Dat-Cre in other previous experiments, to corroborated hyperactivity.

Figures

Figure 1. The lettering labels in Figure 1 B need to be resized to avoid confusion between genotypes in the immunohistochemistry image results. Figure 1I is irrelevant, and the space could be better used if supplementary Figures 2C or 2D are provided here instead. Data in Figure 1J should be separated to allow a comparison of open vs. closed arms entries by genotype.

Figure 2. The variability in striatal dopamine in Selenotf/f; Dat-Cre mice is interesting since other striatal neurotransmitter content seems more consistent across genotypes. The diagram in Figure 2G indicates the microdialysis probe was placed in the dorsal striatum. Could a regional (ventral vs. dorsal striatum) explain the apparently contradictory results between dopamine content in striatal tissue and microdialysis dopamine quantification?

Figure 3. How many neurons were recorded in Selenotf/f; Dat-cre mice for figures 3 B-D, 26, or 27? The difference in sEPSC recorded in striatal cells from Selenotf/f; Dat-cre mice seems driven by only two data points in graph F.

Figure 4. DAT staining in the pictures given in Figure 4C does not show a striking 50% plus difference in DAT expression, as reported in the graphs in Figure 4D or the western blot analysis results in Figures 4A, 4B, and 4C. How do the authors explain that the number of TH+ neurons was significantly reduced in SN and VTA of Selenotf/f; Dat-Cre mice (Figures 4I and 4J), but the average fluorescence intensity of TH is not (Figures 4K and 4L)? DAT immunolabeling in Figure 4K seems unspecific, and some cells do not express TH; what is the author's explanation for this?

Figure 5. The initial hyperactivity in Selenotf/f; Nestin-Cre mice shown in Figure 5B seems to be transient and disappeared after an hour.

Figure 9. Could the effects of amphetamine and methylphenidate at the higher doses produce an increase in stereotypic behaviors that do not necessarily translate to distance traveled?

** As a service to authors, EMBO Press provides authors with the possibility to transfer a manuscript that one journal cannot offer to publish to another EMBO publication or the open access journal Life Science Alliance launched in partnership between EMBO Press, Rockefeller University Press and Cold Spring Harbor Laboratory Press. The full manuscript and if applicable, reviewers' reports, are automatically sent to the receiving journal to allow for fast handling and a prompt decision on your manuscript. For more details of this service, and to transfer your manuscript please click on Link Not Available. **

Referee #2:

This manuscript from Guo et al asks how the loss of ER protein Selenoprotein T (Selenot) dysregulates dopaminergic function and "ADHD-like" behaviors in male mice. Central Selenot knockout, or Selenot loss in dopamine neurons caused a reduction in dopamine transporter expression (DAT), and a hyperlocomotion and increased stereotypy behavioral phenotype that is typical of DAT knockout mice. The hyperlocomotion phenotype is blocked with higher doses of psychostimulants amphetamine and methylphenidate. In mice lacking Selenot in DAT+ cells, there is increased striatal dopamine metabolites and basal dopamine tone, as well as increased excitatory input onto striatal medium spiny neurons, and reduced expression of midbrain Nr4a2. To look at the molecular mechanisms, the authors use siRNA against Selenot, or Selenot overexpression in HEK293 cells, finding DAT and Nr4a2 expression are linked to the degree of Selenot expression. They find in vitro, that Selenot control over DAT expression is mediated through Nr4a2, as NR4a2 knockdown occluded the effects of Selenot overexpression on DAT mRNA. Using in vitro and ex vivo calcium imaging, they found Selenot expression was linked to cytosolic calcium regulation by SERCA, such that SERCA-inhibition-induced cytosolic calcium was potentiated in Selenot overexpressing HEK293 cells and attenuated in the Selenot knockdown or floxed mouse midbrain. In a series of immunoprecipitation experiments, they confirm Selenot has stronger interaction with SERCA vs IP3 or ryanodine receptors.

This manuscript provides important insight into regulation of dopamine transporter expression, which has implications for a wide variety of neurodegenerative and neuropsychiatric disorders. My primary concerns can be summarized as (1) The organization of the manuscript and figures makes the story difficult to follow (2) The over-reliance on representative images without quantification/insufficient validation (3) No Selenot overexpression in vivo to corroborate that Selenot directly controls DAT expression via Nr4a2, experiments that are necessitated by the selection of HEK293 cells, which typically exhibit very low levels of DAT expression.

Response: Thank you very much for the encouraging comments. We have now reorganized the manuscript and figures as suggested, provided quantifications and validations where needed, and performed in vivo SELENOT overexpression to corroborate the regulation. Our point-by-point response is detailed below.

1. The manuscript is extremely difficult to follow as the authors switch back and forth between in vivo and in vitro experiments in the same figure.

Response: Thank you very much. We have followed all your recommendations and have reorganized the manuscript and figures as outlined below.

(a) I recommend moving Figure 5 (Selenot-flox-Nestin-Cre) to the supplement. Then, the idea that the behavioral effects are neuron specific ("Selenot deficiency in whole brain, ...") can follow the first section in the results that describes the behavioral effects in the Selenot-Flox-DAT-Cre (Selenot^{fl/fl};Dat-cre mice display such ADHD-like...).

Response: The previous "Figure 5 (Selenot-flox-Nestin-Cre)" has been moved to the supplement as new Supplementary Figure 3.

(b) Figure 9, with the behavioral rescue with amph/mph, should be additional panels in Figure 1, with perhaps the lower doses (9ABEF) in the supplement.

Response: The previous Figure 9, which includes higher doses, has been moved to the first figure as new Figure 1, U-X. The section with lower doses has been moved to Supplementary Figure 2, P-S.

(c) New Figure 5 should break out the mouse RNAseq into its own figure, showing the volcano (6D), the GO analysis (7A), the heatmaps (6E), the ADHD-associated gene violin plots (6F), and the protein-level validation of DAT and Nr4a2 (6GHI)

Response: New Figure 5 has been created as you suggested, except for the DAT protein, which remains in Figure 4 to maintain the narrative flow of the study.

(d) New Figure 6 becomes the Selenot knockdown and overexpression, and Nr4a2 knockdown occlusion experiment, all in HEK293

Response: New Figure 6 has been revised as suggested, with the addition of protein levels of DAT and NURR1 in the AAV-SELENOT infected mice.

(e) New Figure 7 would better be organized as the HEK293 experiments (knockdown, overexpression, CPA-dependent transcription), then the slice experiments.

Response: New Figure 7 has been revised as suggested.

(f) New Figure 8 would remove panel D with the RNAseq counts from the mice, leaving the figure all HEK293.

Response: New Figure 8 has been reorganized as suggested.

2. The following quantifications or validations are missing or inadequate:

Response: Our responses are detailed below.

(a) Selenot mRNA for siSELT1, siSELT2 in HEK293

Response: *SELENOT* mRNA quantifications for siSELT1 and siSELT2 in HEK293 cells have been added (left panel of Figure 6C). The mRNA for pMyc-SELT was not quantified, as it provides little relevant information for plasmid transfection.

(b) Selenot protein for siSELT1, siSELT2 in HEK293- there's only representative blot in 6A

Response: *SELENOT* protein quantifications for siSELT1 and siSELT2 have been added (left panel of Figure 6B). The same is true for pMyc-SELT (right panel of Figure 6B).

(c) Quantification of the IP experiments in Fig 8FG-there's only representative blots

Response: Quantification of the IP experiments have been added (new Figure 8, E-J).

(d) Quantification of the colocalization in Fig 8HI-there's only representative IHC

Response: Quantification of the colocalization images have been added (new Figure 8, K-N).

(e) Selenot mRNA/protein in the Selenot-flox-DAT-cre animals. There are RNAseq counts in the supplement, but it is unclear if they are different, and they may not be due to bulk RNAseq. An acceptable means of showing decreased Selenot expression in the Selenot-flox-DAT-cre animals would be a quantification of Selenot positive cells as a percentage of TH positive cells-- Figure 1B with TH/Selenot colocalization.

Response: Thank you for the comments. We have performed the quantifications as suggested, expressing the results as the percentage of SELENOT and TH double-positive cells relative to TH-positive cells. The results showed no colocalized staining in *Selenot^{flox};DAT-Cre* mice (new Figure 1, B and C).

(f) unquantified westerns for Selenot-flox-GFAP-Cre and Selenot-flox-Nestin-Cre. Also, region for the western in Supp 5B is unspecified.

Response: The quantifications for Western blots of *Selenot^{flox};Nestin-Cre* (new Supplementary Figure 3C) and *Selenot^{flox};GFAP-Cre* (new Supplementary Figure 5B) mice have been added. We have specified in the figure legend that the analysis was performed “using the whole brain”.

(g) Figure 4K-L. The more appropriate quantification would be number of TH and/or DAT positive nuclei as a percentage of the total nuclei. Fluorescence intensity in IHC is an unreliable indicator of degree of expression.

Response: Thank you for the suggestion. We realize that we did not clearly explain the purpose of Figure 4K-L. The intention was to determine whether the mild reduction of TH levels in the substantia nigra and VTA, as shown in Western blot, was due to a reduction in dopaminergic cell number or a decrease in intracellular TH expression. Because the DAT showed nonspecific immunofluorescence labeling, as noted by Referee #3, we have removed the DAT data and re-quantified intracellular TH expression by normalizing to nuclei intensity. We have also included an explanation of the purpose, stating that, “These results suggest that the mild reduction of TH levels observed in the Western blot analyses was due to a slight reduction in dopaminergic cell number, rather than a change in intracellular TH expression.” (Lines 211-214).

(i) Figure 7J-K. Fluorescence intensity can change as a function of sensor expression. There is no indication that GCaMP is similarly expressed in the Selenot-flox-DAT-cre vs DAT-cre, rendering the “basal fluorescence density” in 7K uninterpretable. It is also worth mentioning the methodologic details regarding the calcium imaging is lacking-e.g. there are no excitation/emission filters, the analysis is insufficiently described (“with ImageJ”).

Response: Thank you for the comments. We have now used the expression of Tdtomato to normalize GCaMP expression (new Figure 7, L-M). Additional methodological details have been included: “A 488 nm and a 561 nm laser were used to excite GCaMP6s and dTomato, respectively. The region of interest (ROI) was defined before quantifying features. Using Time Series Analyzer in Image J software, ROIs were drawn around individual cells, and the fluorescence intensities of the ROIs in the recorded video images were automatically quantified. Data were then acquired and statistically analyzed.” (Lines 605-610).

3. The idea that Selenot directly controls DAT expression via Nr4a2 and SERCA is inadequately supported.

Response: Thank you for the comments. Our responses are detailed below.

(a) HEK293 cells do not normally express high levels of DAT, and are a poor model for neurons as they are normally electrically unexcitable cells. While many of the siRNA knockdown in HEK293 cells recapitulate the knockout in DAT cells (gene and protein expression, reduced CPA-induced Ca²⁺ influx), there are no experiments showing the inverse—that increasing Selenot in vivo also increases Nr4a2 and DAT expression. This is important since the mice are lacking Selenot in DAT cells from development, and the decrease in DAT and Nr4a2 could arise from some other compensatory mechanism. Viral-mediated Selenot overexpression in vivo should increase Nr4a2 and DAT expression as it does in the in vitro experiments.

Response: As you suggested, we performed virus-mediated SELENOT overexpression in

vivo. The new results showed that the *in vivo* overexpression of SELENOT rescued the hyperactivity of Selenot^{fl/fl};Dat-Cre mice (Figure 1, Y-AB), and increased the expression levels of Nr4a2/NURR1 and DAT in the substantia nigra and/or striatum (Figure 6, I-L).

(b) Figure 7M-N show that SERCA influences DAT and Nr4a2 expression, not that this is mediated by Selenot. To make that claim, the authors must repeat the CPA-stimulated DAT and Nr4a2 expression in the Selenot knockdown and/or overexpressing cells.

Response: We have added an experiment to analyze the CPA-stimulated DAT and Nr4a2/NURR1 expression in *Selenot* knockdown cells, and have described in the text that "..., while *SELENOT* pre-knockdown blunted the CPA-stimulated *DAT* and *NURR1* expression (Figure 7K)." (Lines 302-304).

Other concerns

1. Figure 3--the striatal medium spiny neurons have increased excitatory input (greater sEPSC frequency), not increased excitability--to claim increased excitability the authors need to measure rheobase and/or current evoked spikes.

Response: Thank you for the suggestion. We have changed the expression to "increased excitatory input" (Lines 166-167, 189, 339, and 876).

2. Figure 4-- Drd antibodies are notoriously non-specific (see PMID: 2831602). While the Drd2 antibody from Abcam used in this manuscript has been validated with knockout animals, there is no knockout validation for the Drd1 antibody from santa-cruz. Previous versions of santa-cruz Drd antibodies continue to show signal in western blot in knockout animals. If the authors cannot find a citation showing this antibody is specific, the band for Drd1 should be removed from the manuscript so as not to perpetuate the flawed rationale that an antibody is ok to use since it's been used in another manuscript.

Response: Thank you for the advice. We have another Drd1 antibody from Proteintech (Cat # 17934-1-AP), but we could not find a citation validating its use in knockout animals. Therefore, as you suggested, we have removed the band for Drd1/D1R.

3. The manuscript seems to have been conducted exclusively in male mice. This should be specified in the abstract.

Response: We have specified male mice in the abstract.

4. How are "social recognition index" and "social preference index" calculated?

Response: We have added the calculation to the methods section: "A social preference index was calculated as the ratio of time spent exploring the stranger 1 cage to the total time spent exploring both stranger 1 and the empty cages. A social recognition index was calculated as the ratio of time spent exploring the stranger 2 cage to the total time spent exploring both stranger 1 and stranger 2 cages." (Lines 494-498).

5. Lines 107-112. "suggesting accelerated dopamine metabolism..." ... "indicate that dopamine metabolism was impaired". These two statements contradict one another. Based on the increased dopamine metabolites and MAO-A expression, dopamine metabolism appears to be increased in the selenot-flox-DAT-cre mice. The microdialysis seems to indicate a greater

basal dopamine tone in the selenot-flox-DAT-cre mice, which is not explained by increased midbrain dopamine neuron firing, but does make sense given the loss of DAT.

Response: We apologize for any confusion caused by our wording. We considered that the acceleration could also be viewed as a form of impairment, which is why we initially used “impaired”. Based on your feedback, we have rephrased the sentence to: “These results indicate increased dopamine metabolism and a greater basal dopamine tone in the striatum of *Selenot^{f/f};Dat-cre* mice.” (Lines 177-178). Thank you.

6. There are instances in which unpaired t-tests are used for paired measurements-
Supplemental 2J, 2N, 5L,5O

Response: Thank you for bringing this to our attention. We have corrected the analysis to use paired t-tests for these measurements (as shown in Supplementary Figure 2, J and N, and Supplementary Figure 4, J and M in the revision).

Minor concerns

1. In the abstract, specify that it's the number of dopamine neurons in the SNc that are reduced.

Response: As suggested, we have specified that “the number of dopaminergic neurons in the substantia nigra were slightly reduced”. (Lines 26-27).

2. Figure 1L heatmap for the selenot-flox-dat-cre, mice appear to spend less time interacting with objects in general. The exploration time in both genotypes should be reported in the supplement, not just the ratio of novel to familiar.

Response: We have replaced the previous heatmaps with more representative ones (new Figure 1O). As suggested, we have also provided the exploration time for both genotypes in the supplement (new Supplementary Figure 2, C and D).

3. Figure 4-what is the arrow on the VStr VMAT2 western?

Response: The arrow indicates the band for VMAT2. We have also added an arrow on the dorsal striatum, and explained in the figure legend (Lines 895).

4.The violin plots of "adhd candidate genes" - it's not really clear from the plots that they are significant from each other and to what extent these represent a large percentage of the candidate genes-i.e. how many ADHD candidate genes are there per DisGenet?

Response: We have converted the violin plots into a table with fold change and *P* values (new Figure 5E). These genes are DEGs from the RNA-seq and represent only a small portion of the total candidate genes. We have noted in the figure legend that “List of ADHD candidate DEGs identified by DisGenet. This database contains a total of 842 genes.” (Lines 916-917).

5. Many of the RNAseq visualizations would be more informative if there was a supplemental table of DEGs showing logFC and p-value. The volcano plot is fine, but only two labeled genes, which are then repeated in the adjacent heatmap, isn't very informative.

Response: As suggested, we have provided a supplementary table of DEGs showing logFC and *P* value (new Supplementary Table 2). Thank you for the suggestion.

Referee #3:

General Summary: In their manuscript, Guo et al. presented an extensive study on the role of selenoprotein T (Selenot) on dopamine neuron function, the regulation of DAT through the transcription factor NURR1 (a mechanism involving Ca^{+2} influx to the ER through the SERCA2 pump) and a putative role in attention deficit and hyperactivity disorder (ADHD)-like behaviors. The authors used a Selenot^{fl/fl}; Dat-Cre mouse model for most of their experiments and performed behavioral, electrophysiological (patch-clamp and EEG), HPLC with mass spectrometry, immunohistochemistry and western blot experiments to determine SELENOT role in striatal and midbrain dopamine function. They further tested whether their observations on dopamine function were due to an effect of Selenot deletion on neuronal or astrocytes populations comparing a Selenot^{fl/fl}; Nestin-cre and Selenot^{fl/fl}; Gfap-cre mice strains, respectively. Furthermore, Guo et al., performed RNA seq and qPCR experiments in substantia nigra of Selenot^{fl/fl}; Dat-Cre mice and identified Nurr1 to be downregulated along with Dat. Then, to determine a possible mechanism of SELENOT on DAT expression, they used transfection in HEK293 cultures, calcium imaging and co-immunoprecipitation experiments and determine that SELENOT interact with SERCA2 channels to modulate cytoplasmic Ca^{+} and the expression of NURR1 and DAT. Finally, the authors rescued the hyperactive phenotype observed in Selenot^{fl/fl}; Dat-Cre mice using amphetamine or methylphenidate treatment. Guo et al., concluded that Selenot in dopaminergic neurons maintains proper dopamine signaling and protects against ADHD-like behaviors.

Opinion: Guo et al., did a great deal of work in this manuscript. However, the link between the putative ADHD-like behaviors on their mouse model and the findings on the interaction of SELENOT with SERCA and Ca^{2+} , documented in HEK293 cell cultures, is directly supported only by the data showed in Figure 7J-K.

Response: Thank you very much for your summary. We have performed new *in vivo* experiments, as addressed in the point-by-point response below, to consolidate the link. As a reminder, the figures and the corresponding order of results description have been significantly rearranged based on the feedback from Referee #2.

Primary concern: Although this is an extensive study involving several methodologies and techniques, it does not make a cohesive argument to support that SELENOT is indispensable in protecting against dopamine dysfunction and attention deficit and hyperactivity disorder (ADHD)-like behaviors. First, the authors have similar behavioral observations using Selenot^{fl/fl}; Dat-Cre mice and the less dopamine neuronal-specific Selenot^{fl/fl}; Nestin-Cre mice. There is also the possibility that the hyperactivity and DAT expression anomalies reported in the Selenot^{fl/fl}; Dat-Cre mice are related to the Dat-Cre genotype, even as heterozygous. Please see the comment in the Materials and Methods section below and Machado-Acosta et al. 2021, doi: 10.1038/s41598-021-82600-x).

Response: Thank you for the comments. The detailed answers have been provided in the

response to your individual comments. For your convenience, here is the index: 1) Our rationale for why the similar behavioral observations in *Selenot^{f/f};Dat-cre* and *Selenot^{f/f};Nestin-cre* mice are supportive has been explained to your “Lines 68-70” comment. 2) Our strain of *Dat-cre* mice is different from the one referenced in the DOI you provided. Information about the difference can be found on the JAX website. Additionally, we have now included both DAT expression and locomotion activity data, directly comparing *Selenot^{f/f};Dat-cre* to *Dat-cre* mice. Details are provided in the response to your “Lines 397-403” comment.

Minor concern: The manuscript is in general well-written; however, some parts will require proper grammar revision.

Response: Thank you. We have asked a professor in the U.S. to help revise the language.

Other specific comments and questions follow below:

Title

I am unsure if the title conveys what the authors demonstrated with their *Selenotf/f; Dat-Cre* mouse as a model of ADHD-like behaviors. The author provided evidence to support the interaction of SELENOT with SERCA2 and this could in turn modulate NURR1 and DAT expression, but there is no direct evidence that this mechanism is responsible for ADHD-like behaviors observed in *Selenotf/f; Dat-Cre* mouse.

Response: We have now added new experiments by *in vivo* overexpressing SELENOT in dopaminergic neurons using a TH promoter-initiated viral vector. These experiments showed that the overexpression rescued the hyperactivity in *Selenot^{f/f};Dat-cre* mice (Figure 1, Y-AB) and upregulated the expression of NURR1 and DAT in the midbrain (Figure 6, I-L). We also demonstrated that SELENOT knockdown blunted the effect of CPA on DAT and NURR1 expression in cells (Figure 7K). We hope you agree that the title is reasonable in light of this new evidence.

Abstract

Lines 28-29. The study seems to have dropped the finding of increased monoamine oxidase A in relation to ADHD-like behavior in *Selenotf/f; Dat-Cre* mice. They discussed this in lines 362-367, but it could be removed from the abstract since this was not further studied in the present manuscript.

Response: As suggested, we have removed the mention of increased monoamine oxidase A in the abstract.

Introduction

Lines 68-70. This statement is an overstatement since the authors did not provide direct evidence that *Selenot* deficit only in dopaminergic neurons produced increased hyperactivity. They provided supporting evidence that SELENOT mediated Ca²⁺ homeostasis in HEK293 cells. *Selenotf/f; Nestin-Cre* mice also developed similar hyperactivity behaviors.

Response: First, please allow us to explain the logic in response to your comment on “Lines 161-190”. If *Selenot^{f/f};Dat-Cre* represents A (*Selenot* deficit in dopaminergic neurons), and

Selenot^{fl/fl};Nestin-Cre represents A+B+C+D (*Selenot* deficit in many types of neurons), with the phenotype being X. Our results show that A causes X and A+B+C+D also causes X, which logically implies that A is the major, if not the only, contributor to X. Moreover, A+B+C+D leads to a milder, not more severe, phenotype X compared with A alone, which suggests that B+C+D are not positive contributors to X, but are more likely involved in a compensatory circuit that alleviates the severity of X.

Importantly, we have now added *in vivo* evidence demonstrating that TH-promotor-driven SELENOT overexpression in dopaminergic neurons rescued the hyperactivity in *Selenot^{fl/fl};Dat-cre* mice (Figure 1, Y-AB) and upregulated the expression of NURR1 and DAT in the midbrain (Figure 6, I-L). Regarding Ca²⁺ regulation, we provide a variety of supporting evidence, including *in vivo* data from GCaMP viral vectors showing SELENOT-mediated Ca²⁺ homeostasis (as Figure 7, L-N in the resubmission), *in vitro* evidence in cells (Figure 7), and biochemical evidence of protein interactions and colocalization (Figure 8).

We hope you would accept our reasoning and find our statement to be appropriately supported, especially the claim: “In this study, we demonstrated the essential roles of dopaminergic neuron SELENOT in preventing ADHD-like behaviors. The underlying mechanisms involved the SELENOT-mediated nigral ER-cytosol Ca²⁺ homeostasis, dopamine transporter (DAT), and striatal dopamine transmission.”. Furthermore, we did not conclude that “*Selenot* deficit only in dopaminergic neurons produced increased hyperactivity” in this sentence. Thank you very much.

Results

Lines 78-79. What is the percentage of colocalization between SELENOT and TH? How substantia nigra (SN) expression of SELENOT is compared to dopamine neurons in the ventral tegmental area (VTA)?

Response: We have added the analysis of the colocalization between SELENOT and TH, as well as the expression differences between the substantia nigra and VTA. We state in the text that “SELENOT was present nearly in all dopaminergic neurons and showed no expression difference between the substantia nigra and VTA (Figure 1, B-D).” (Lines 84-85)

Lines 85-86. The authors report no differences in central open-field crossings, but the activity heatmap examples in Figure 1D, could suggest an increase in crossings in *Selenot^{fl/fl};Dat-cre* mice compared with *Selenot^{fl/fl}* mice. Maybe a better example that reflects the group average is needed in this figure?

Response: Thank you very much for the suggestion. We have replaced the previous heatmaps with more representative ones to reflect the group average (Figure 1F in the revision).

Lines 88-89. Could the increase in open arm entries indicate increased impulsivity or just reflect the hyperactivity in *Selenot^{fl/fl};Nestin-Cre* mice?

Response: Thank you for the comments. We believe you meant *Selenot^{fl/fl};Dat-Cre* mice, as the content in these lines refer to this genotype. You raised an interesting point. Generally speaking, open arm entry is used to evaluate anxiety, but it is not sufficient to assess impulsivity. Instead, the cliff avoidance reaction test is used to evaluate impulsivity by

counting the frequency of a mouse falling from the platform. Reaching the platform edge is not considered impulsivity.

Lines 90-92. The novel object recognition test results can be interpreted as a decrease in object exploration (object avoidance) when looking at the provided heat maps at the 24 hours test in Figure 1N. Selenotf/f; Nestin-Cre mice seem to spend more time away from the objects and in the corners of the test chamber.

Response: Thank you for the comments. We have now included the total exploration time (new Supplementary Figure 2, C and D). As you can see, total exploration time may vary among different animals. The preference is calculated as a paired percentage within the same animal, so the results should theoretically not be affected by total object exploration time. We have also updated our analysis by excluding mice with a total exploration time of less than 5 seconds (Lines 505-506), following consultation with colleagues.

Lines 92-93 and 96-97. In which context and how repetitive behaviors reported here and in Figure 1P were measured?

Response: Sorry we missed this part. We have added the following to the Methods section: "For the repetitive behavior test, the mouse was placed in a plastic cage containing 15 cm of corn cob bedding with a lid and recorded for its behavior for 20 min. The incidences of rearing, digging, tail flicking, climbing, and jumping were counted." (Lines 507-510).

Lines 97-98. The heat map provided in Supplementary Figure 2I could suggest that Selenotf/fl; Dat-Cre mice have higher social interaction time than Selenotf/fl mice.

Response: We have replaced them with more representative heatmaps (new Supplementary Figure 2I). Thank you.

Lines 105-111. How do the authors explain increased striatal DA in the micro-dialysis experiments but not in DA tissue content? Do Selenotf/f; Dat-Cre mice have an accelerated or impaired dopamine metabolism? The author uses both statements in different places in the paragraph.

Response: The increased extrasynaptic dopamine observed in the microdialysis experiment is consistent with the reduced expression of DAT, which normally transports extrasynaptic dopamine back into presynaptic terminals. Therefore, it does not contradict the normal dopamine levels in whole tissue. The use of "accelerated or impaired" was a language issue on our part. We initially thought that acceleration could also be considered a form of damage relative to the physiological condition. We have now revised the text to use "accelerated" or "acceleration" consistently.

Lines 111-113. The microdialysis technique does not have enough resolution to support the statement of a "high level of dopamine level retained in the synaptic gap in the striatum of Selenotf/f; Dat-Cre mice."

Response: In response to your comment, we attempted to use fast-scan cyclic voltammetry (FSCV) to measure neurotransmitters with equipment from our core facility that had not been used previously, and we purchased a new set of carbon-fiber microelectrodes. However, the

technique ultimately did not work, even with the help of an engineer. While microdialysis remains the most widely used method to measure extrasynaptic materials, the phrase “in the synaptic gap” may have been overstated. We have revised this expression and updated the statement to: “These results indicate increased dopamine metabolism and a greater basal dopamine tone in the striatum of *Selenot^{fl/fl};Dat-cre* mice.” (Lines 177-178) as suggested by Referee #2. Thank you.

Line 113. Could the increase in glutamic acid indicate tissue damage (i.e. glutamate exocytosis)?

Response: Neurotransmitters were measured in the striatum. We believe the possibility of tissue damage in the striatum is very low, as most neurotransmitter levels remained normal, and the knockout occurred in the substantia nigra. Even in this region, we did not observe ER stress or apoptosis, as shown in Supplementary Figure 8, F and G).

Lines 121-123. How were recordings on sEPSC in medium spiny neurons performed? What was the location and type of medium spiny neuron recorder? There is no description of this in the methods section.

Response: Sorry we missed this part. We have now included this in the Methods section: “... and striatal neurons in the dorsal striatum were also recorded, with most of these neurons being medium spiny neurons by morphology. However, we did not further distinguish them as D1R-positive or D2R-positive neurons.” (Lines 581-584).

Lines 123-124. Are the authors suggesting increased striatal extrasynaptic dopamine directly increased sEPSC in medium spiny neurons?

Response: We may have slightly overstated this. We have now revised it to: “This result suggests enhanced postsynaptic excitatory input and is consistent with the increased abundance of extrasynaptic dopamine in the striatum of *Selenot^{fl/fl};Dat-cre* mice.” (Lines 188-190).

Lines 134-145. Do the authors observe any regional differences in markers of DAT or TH expression between the substantia nigra vs. the ventral tegmental area or dorsal vs. ventral striatum within genotypes?

Response: Based on the observations in Figure 4, C-F, I and J, we could not find an apparent difference in TH or DAT expression between the substantia nigra and the VTA, or between the dorsal and ventral striatum within genotypes. Further subdivision may be needed to achieve clear resolution in the comparison, such as examining medial/lateral regions or specific loci like the nucleus accumbens.

Lines 141-145. Quantification by Western blots is usually considered a better parameter than fluorescence intensity from immunolabeling, particularly to assess changes in protein expression. One reason is unspecific fluorescence labeling, like the one visible in Figure 4K for DAT, which does not seem to correspond to TH-positive cells.

Response: We did not clearly explain the purpose of the experiment: it was to determine whether the mild reduction in TH observed in the Western blot was due to a decrease in

dopaminergic cell number or a reduction in intracellular TH expression, and also to assess intracellular DAT expression. We have now removed the DAT staining due to the nonspecific immunofluorescence you noted, and re-quantified intracellular TH expression by normalizing to DAPI. Additionally, we have added a summary sentence to clarify the findings: “These results suggest that the mild reduction of TH levels observed in the Western blot analyses was due to a slight reduction in dopaminergic cell number, rather than a change in intracellular TH expression.” (Lines 211-214).

Lines 145-147. Which region of the striatum was used for DAT-Cre vs. Selenotf/f comparison? The blots provided in Figure 4A show that the ventral striatum has higher DAT expression than the dorsal striatum in both Selenotf/f and Selenotf/f; DAT-Cre mice. Also, β -actin expression in Supplementary Figure 4C seems lower than in blots provided in other figures.

Response: We used the whole striatum, as noted in the figure legend (Supplementary Figure 8, A and B). Bands from two separate blots should not be compared. Therefore, we cannot determine whether DAT expression is higher in the dorsal striatum based on these two gels. However, as previously addressed in response to your comment on “Lines 134-145”, DAT expression was not found to differ between the dorsal and ventral striatum. Regarding the β -actin question, we checked whether films with exposures matching the other blots were available, and similarly checked for other proteins. Two of these proteins, including β -actin (Supplementary Figure 8A) and VMAT2 in Figure 4A, have now been included with heavier exposures.

Lines 161-190. Results showed in Figure 5 related to the behavioral testing, DAT and TH protein expression quantification in substantia nigra and striatum from Selenotf/f; Nestin-Cre mice suggest that the dopamine deficits observed in Selenotf/f; Dat-Cre may not be specific to dopamine circuitry. The authors showed that a similar effect can be achieved with an unspecific neuronal deficit of Selenot.

Response: Thank you for the comments. Please allow us to explain the logic: if *Selenot^{f/f};Dat-Cre* represents A (*Selenot* deficit in dopaminergic neurons), *Selenot^{f/f};Nestin-Cre* represents A+B+C+D (*Selenot* deficit in many types of neurons), with the phenotype being X, our results show that A causes X and A+B+C+D also causes X. This logically indicates that A is a major, if not the only, contributor to X. Moreover, compared with A alone, A+B+C+D causes a milder, but not more severe, phenotype X, which suggests that B+C+D are not positive contributors to X but are more likely involved in a compensatory circuit that alleviates the severity of X. We hope that you find our explanation acceptable. Thank you.

Lines 180-182. In the social preference and social recognition tests, the position of the stranger mice shown in the heat maps of Supplementary Figures 5K and 5N is different from that in the experiments for Selenotf/f; Dat-Cre (Supplementary Figures 2I and 2M).

Response: Yes, this was an unintentional inconsistency. The cages were placed in ipsilateral corners for *Selenot^{f/f};Dat-Cre* mice and in contralateral corners for *Selenot^{f/f};Nestin-Cre* mice. Although both placements are acceptable since the distance to the door is the same, we will ensure consistency in future studies. We have added a note to the legend (new

Supplementary Figure 4): “Note: the cages were unintentionally placed in contralateral corners for this genotype”. Thank you.

Lines 203-207. These results could predict that overexpression of SELENOT can rescue the hyperactive phenotype observed in *Selenotf/f; Dat-Cre* and *Selenotf/f; Nestin-Cre* mice lines.

Response: As you indicated, we have now included this experiment using *Selenot^{fl/fl};Dat-cre* mice, as stated in the text: “Furthermore, to determine whether recovering SELENOT expression rescues the hyperactivity observed in *Selenot^{fl/fl};Dat-cre* mice, we *in vivo* overexpressed SELENOT in dopaminergic neurons of the substantia nigra using a TH promoter-initiated viral vector (Figure 1, Y and Z). Similar to the effect of amphetamine, SELENOT overexpression did not appear to affect locomotion in *Selenot^{fl/fl}* control mice. In contrast, the overexpression successfully rescued the hyperlocomotion phenotype in *Selenot^{fl/fl};Dat-cre* mice (Figure 1, AA and AB).” (Lines 121-127).

Line 207. Why the authors selected substantia nigra for the RNA-seq experiments? In *Selenotf/f; Dat-Cre*, DAT was more drastically reduced in striatal tissue. Striatal RNAseq analysis could also provide supporting evidence to corroborate the changes in dopamine metabolism reported in Figure 2F.

Response: RNAs are presumably located in neuronal cell bodies, including the nucleus and surrounding cytoplasm, but are unlikely to be found in the projected terminals. For dopaminergic neurons, the cell bodies are located in the substantia nigra, while only the fibers and terminals are present in the striatum. This is why we chose the substantia nigra for RNA-seq.

Lines 249 and 251. Could CPA treatment rescue the phenotype of *Selenotf/f; Dat-Cre* mice?

Response: CPA is a mycotoxin, inducing various toxic symptoms, including in the nervous system (ref: Nishie K, Cole RJ, Dorner JW (1985) Toxicity and neuropharmacology of cyclopiazonic acid. *Food Chem Toxicol* 23: 831-839). Therefore, we believe that CPA is not suitable for use *in vivo* for this purpose.

Lines 261-264. Were *Serca1/Atp2a1*, *Ip3r1* and *Ryr2* higher in the substantia nigra of *Selenotf/f; Dat-Cre* mice compared with *Selenotf/f* mice? Otherwise, this could suggest that the *Selenotf/f* genetic modification alone modify the expression of these genes and explain the behavioral observations on *Selenotf/f; Nestin-Cre* mice.

Response: No, the expression levels are not higher in the substantia nigra of *Selenot^{fl/fl};Dat-cre* compared with *Selenot^{fl/fl}* mice. We have clarified in the substantia nigra RNA-seq section that, “Expression levels of the ER-associated calcium channel genes, including *Serca1/2/3* (also named as *Atp2a1/2/3*), *Ip3r1/2/3*, and *Ryr1/2/3*, were not significantly different between *Selenot^{fl/fl};Dat-cre* and *Selenot^{fl/fl}* mice.. (Figure 5D) ... (Supplementary Figure 9, B and C).” (Lines 257-264).

Materials and Methods

Lines 397-403. I cannot find whether the *Dat-Cre*, *Nestin-Cre*, or *Gfap-Cre* genotype was bred to generate homozygous or heterozygous mice. Can the authors provide more details on

the source of the Cre lines? It will be essential to know how the authors developed a DAT-Cre line without significantly altering DAT expression. This is important since Dat-Cre mice lines develop hyperactivity and attenuated hyperlocomotion in response to amphetamine and lower DAT expression, even as heterozygous (See, for example, Machado-Acosta et al. 2021, doi: 10.1038/s41598-021-82600-x).

Response: 1) The Cre genotypes were bred to be heterozygous, as described in our text: “The Cre lines were first crossbred with *Selenot^{fl/fl}* mice to generate *Selenot^{fl/fl};Dat-cre*, *Selenot^{fl/fl};Nestin-cre*, and *Selenot^{fl/fl};Gfap-cre* mice. These mouse lines were then re-bred with *Selenot^{fl/fl}* mice to produce littermates with an identical genetic background.”. A new statement has now been followed that “As such, the Cre genotypes in the conditional knockout lines were all heterozygous.” (Lines 457-461). 2) We have added the sources of the Cre lines (Lines 452-454). The Dat-cre line was obtained from Jackson Laboratory (Strain # 006660) and imported through Cyagen. The Nestin-cre and Gfap-cre lines were purchased from Cyagen (Cat #C001446 and C001062, respectively). 3) The DAT-cre line in the study by Kauê Machado Costa et al is strain #020080 from Jackson Laboratory. As stated on the JAX website (<https://www.jax.org/strain/006660> and <https://www.jax.org/strain/020080>), the strain #006660 is “...without disrupting endogenous dopamine transporter expression”, and the strain #020080 “...was designed to both abolish DAT gene function and place Cre recombinase expression...”. Therefore, the observations by Kauê Machado Costa et al. are not surprising. In addition, we have verified that DAT expression was not different in the substantia nigra or whole striatum between the *Dat-cre* line and *Selenot^{fl/fl}* mice (Supplementary Figure 8, A-B). To further address your concern, we independently performed behavioral tests on *Selenot^{fl/fl};Dat-cre* and *Dat-cre* mice, and “results from the open-field test still suggested that *Selenot^{fl/fl};Dat-cre* mice were hyperactive compared to the *Dat-cre* control mice (Supplementary Figure 8, C-E).” (Lines 225-227).

Lines 543-547. What was the read depth on the RNAseq? I read the reference provided but cannot find the read depth there either.

Response: We double-checked with the company technician, and there is no specific parameter for read depth in RNA-seq. Instead, we have provided a table with detailed quality control information for the RNA-seq and state in the text: “The number of valid bases after quality control in each sample exceeded 6.4G. Both Q20 and Q30 values for each sample were over 98%, and no GC bias was found (Supplementary Table 1).” (Lines 245-246).

Line 521. I understand the rationale to use DAT-Cre mice as a control in this experiment but also highlight the importance of comparing DAT-Cre mice with the *Selenot^{fl/fl};Dat-Cre* in other previous experiments, to corroborated hyperactivity.

Response: As detailed in the response above, we compared *Dat-Cre* mice with *Selenot^{fl/fl};Dat-cre* mice. The results showed that DAT expression was not different between these two genotypes, and *Selenot^{fl/fl};Dat-cre* mice remained hyperactive (new Supplementary Figure 8, A-E).

Figures

Figure 1. The lettering labels in Figure 1 B need to be resized to avoid confusion between

genotypes in the immunohistochemistry image results. Figure 1I is irrelevant, and the space could be better used if supplementary Figures 2C or 2D are provided here instead. Data in Figure 1J should be separated to allow a comparison of open vs. closed arms entries by genotype.

Response: As suggested, we have rearranged Figure 1B to avoid confusion by reducing the size of the genotype labels and increasing the spacing between them. Figure 1I has been removed, and previous Supplementary Figures 2C and 2D are now provided as new Figure 1L and 1M. Additionally, Figure 1J (now Figure 1K in the revision) has been separated into total entries, open arm entries, and closed arm entries.

Figure 2. The variability in striatal dopamine in *Selenotf/f; Dat-Cre* mice is interesting since other striatal neurotransmitter content seems more consistent across genotypes. The diagram in Figure 2G indicates the microdialysis probe was placed in the dorsal striatum. Could a regional (ventral vs. dorsal striatum) explain the apparently contradictory results between dopamine content in striatal tissue and microdialysis dopamine quantification?

Response: Microdialysis measures extrasynaptic dopamine, while striatal tissue measures total dopamine. These two results are not contradictory. The increased dopamine measured by microdialysis is consistent with the reduced expression of DAT, which normally transports extrasynaptic dopamine back into neurons. In addition, we corrected the error bar in Figure 2A from SD to SEM.

Figure 3. How many neurons were recorded in *Selenotf/f; Dat-cre* mice for figures 3 B-D, 26, or 27? The difference in sEPSC recorded in striatal cells from *Selenotf/f; Dat-cre* mice seems driven by only two data points in graph F.

Response: We have reviewed the data, and the correct number should be 26. We understand the second point, but those two data points are not statistical outliers. Additionally, if we look at it this perspective, there are also two high data points in the *Selenot^{f/f}* control group. Therefore, the difference should remain affirmative. Thank you.

Figure 4. DAT staining in the pictures given in Figure 4C does not show a striking 50% plus difference in DAT expression, as reported in the graphs in Figure 4D or the western blot analysis results in Figures 4A, 4B, and 4C. How do the authors explain that the number of TH+ neurons was significantly reduced in SN and VTA of *Selenotf/f; Dat-Cre* mice (Figures 4I and 4J), but the average fluorescence intensity of TH is not (Figures 4K and 4L)? DAT immunolabeling in Figure 4K seems unspecific, and some cells do not express TH; what is the author's explanation for this?

Response: 1) We used another software, Image-pro Plus 6.0 for immunohistochemistry (Wang et al, *Survivin expression quantified by Image Pro-Plus compared with visual assessment. Appl Immunohistochem Mol Morphol* 2009, 17: 530-535), as revised in the text (Lines 542-543). The results of quantification showed similar differences (Figure 4C). 2) Regarding the concerns that TH cell number decreased but fluorescence intensity did not, we have stated in the text: "These results suggest that the mild reduction of TH levels observed in the Western blot analyses was due to a slight reduction in dopaminergic cell number, rather than a change in intracellular TH expression." (Lines 211-214). We further discuss: "the

decreased number of dopaminergic neurons is not due to cellular stress but is likely the result of proliferation arrest of progenitors during embryonic development, given that SELENOT can promote G1-to-S cell cycle transition (Shao et al., 2019).” (Lines 236-239). 3) The immunofluorescence for DAT indeed exhibited unspecific and uneven staining. Since this supporting data is rather minor, we have removed the DAT immunofluorescent staining.

Figure 5. The initial hyperactivity in *Selenotf/f; Nestin-Cre* mice shown in Figure 5B seems to be transient and disappeared after an hour.

Response: Thank you for your careful observation. This data has been moved to Supplementary Figure 3F in the resubmission. Yes, it indeed looked that way. This may be due to other mechanisms coming into play after an hour in the whole brain knockout mice. We also observed a subtle difference in repetitive behaviors and a variation in recognition memory between *Selenot^{fl/fl};Dat-cre* and *Selenot^{fl/fl};Nestin-cre* mice. While it is difficult for us to predict the mechanisms at this stage, we have added a note in the results: “Interestingly, the hyperactivity appeared to be transient and disappeared after an hour (Supplementary Figure 3F).” (Lines 141-142). Thank you.

Figure 9. Could the effects of amphetamine and methylphenidate at the higher doses produce an increase in stereotypic behaviors that do not necessarily translate to distance traveled?

Response: Thank you for the comments. This figure has been moved to Figure 1, U-X and Supplementary Figure 2, P-S in the resubmission. If the effects of amphetamine and methylphenidate in *Selenot^{fl/fl};Dat-cre* mice at higher doses were due to an increase in stereotypic behaviors, these effects should also be observed in *Selenot^{fl/fl}* control mice. However, the distance travelled in the control mice was not reduced following treatment with these two drugs.

Dear Dr. Zhu,

Thank you again for resubmitting your revised manuscript EMBOJ-2024-116706R-Q, "SELENOT mediates calcium flux via SERCA2 and maintains dopaminergic DAT against ADHD-like behaviors" to The EMBO Journal for our consideration. It has been sent back to the two original referees that had previously assessed the initial version of your manuscript, and we have now received their comments, which I have already shared with you (they are included again below).

I am very pleased to say that both referees find the revised version of your manuscript significantly improved and the majority of their initially raised criticisms and concerns successfully addressed. They have only a few remaining comments, which I kindly ask you to address in a final revision. Please include in your resubmission a point-by-point response addressing all remaining points and describing any changes to the manuscript.

From the editorial side, there are also a few corrections and changes that we need you to make in this final version of your manuscript before we can proceed with its acceptance for publication. Please include in your resubmission a cover letter detailing how the points below are addressed:

- Please note that all co-corresponding authors must provide valid institutional e-mail addresses as well as ORCID accounts in their profiles in our manuscript handling system. Currently, the co-corresponding author Dr. Xiong Zhang has provided neither an institutional e-mail address nor an ORCID account. Apart from the author's profile in the manuscript tracking system, the institutional e-mail address of Dr. Zhang should also be provided on the title page of the revised manuscript.

- The revised manuscript file should be provided in .docx format.

- At EMBO Press we ask authors to provide source data for the main manuscript Figures. Our source data coordinator has already contacted you separately discussing which Figure panels we would need source data for and also providing you with helpful tips on how to upload and organize the files.

- Please remove the Figures from the main manuscript file. Each Figure should be uploaded to our manuscript tracking system as an individual, high-resolution file. The Figure legends should remain in the manuscript, below the References list. For more information on the preparation of Figures, please see our guide for authors: <https://www.embopress.org/page/journal/14602075/authorguide#figureformat>.

- During our standard pre-acceptance Figure checks, our Data integrity team raised concerns regarding the Western blots presented in your Figures that require clarification before we can accept the manuscript for publication here. In particular:

1. We detected unexpected blot similarity between Appendix Figure 3C and Appendix Figure 9B that is not detailed in the Figure legends. Please double-check these blots and clarify if this is the same experiment; if so, why is it repeated in different Figures? why does the actin control look identical between the two but the SELENOT (Selenot-fl/fl; Selenot-fl/fl; Nestin-Cre) samples do not? why does the actin control look like it was run on a different gel from the samples? is it a loading control for all shown samples? why is processing of the blot images different between the two Figures?
2. We noticed that the actin control shown in Figure 4A has not been run on the same gel as (at least) many or all of the samples shown above; if it is not used as a loading control, why is the actual loading control not shown? how was quantification of the samples' bands performed if there was no loading control? if the included actin control was not meant to be a loading control, what is its purpose? how was this experiment performed (e.g. which bands are cropped from the same gel/run, how many times and for which proteins was the blot stripped/reprobed etc.)?

We kindly request you to:

- i. check again all Western blots shown in the Figures of your manuscript, as well as in your Appendix Figures;
- ii. correct/replace the blots if necessary;
- iii. detail any intentional reuse in the legends of the respective Figures;
- iv. describe in detail in the Methods of your revised manuscript how these experiments were performed;
- v. explain and clarify the issues in detail in your cover letter;
- vi. include in your Source Data the original, uncropped blots (including the molecular weight markers) of all technical replicates for each blot shown in the Figures.

- Please include in your Data availability statement the permanent link (URL) to your deposited RNA-seq data, at which the dataset will be publicly available at the time of publication. We also note that the dataset PRJNA910327 is not publicly available yet. Please make sure that it is either released before resubmission or that a reviewer access code (token) is temporarily provided in the Data availability statement.

- Please change the heading "Competing interests" to "Disclosure and competing interests statement".

- The author contributions statement should be removed from the manuscript file. Instead, we use CRediT to specify the contributions of each author in the journal submission system. Please feel free to use the free text box to provide more detailed

descriptions during submission. See also our guide to authors for more information:
<https://www.embopress.org/page/journal/14602075/authorguide#authorshipguidelines>.

- Please note that all Figure callouts should be listed sequentially in your revised manuscript.
- When you are ready to submit your revision, please also upload a complete author checklist, which you can download from our author guide (<https://www.embopress.org/page/journal/14602075/authorguide>). Please note that the checklist will also be part of the Peer Review File.
- Please include in section "Mice and chemicals" of your Methods the reference number for approval of your experiments involving animals by the authority granting ethics approval.
- The Appendix file needs to be uploaded as a single file in PDF format. Please include the heading "Appendix for" followed by the manuscript's title on its first page. That should be followed by a brief Table of Contents including page numbers listing all items included in the Appendix file. The nomenclature of these items should be "Appendix Figure S#" and "Appendix Table S#" throughout the Appendix file. All callouts in the main manuscript file must be updated accordingly. The information currently provided in the supplementary tables 3 and 4 (information on DNA oligos, siRNA sequences and antibodies) can be incorporated in your Reagents and Tools table (please see next point for more details and instructions).
- Please rename your supplementary table 2 to "Dataset EV1", upload it as a separate Excel file, and update its file name, title, legend, and all callouts throughout the manuscript accordingly. Its caption should be provided in an individual tab/sheet in the same Excel file.
- The materials and methods need to be described in the manuscript using our structured methods format, which is now required for all research articles. According to this format, the Methods section includes a single "Reagents and Tools Table" -listing key reagents, experimental models, software and relevant equipment including their sources and relevant identifiers- followed by a "Methods and Protocols" section describing the methods. Please download and fill our Reagents and Tools Table template (.docx), which you can find in our author guide:
<https://www.embopress.org/page/journal/14602075/authorguide#structuredmethods>. When submitting your revised manuscript, please do not include the Reagents and Tools Table in the Methods section of the manuscript but instead upload it as a separate file choosing the file type "Reagent Table".
- Please note that EMBO press papers are accompanied online by:
 - A) a short (2 sentences) summary of the findings and their significance,
 - B) 2-5 short bullet points highlighting the key results, and
 - C) a synopsis image in .jpg or .png format that is exactly 550 pixels wide and 300-600 pixels high (the height is variable). Please note that the text needs to be legible at the final size.Please upload this information along with your revised manuscript (the text for A and B should be provided in a separate Word file).
- During our routine pre-acceptance checks, our data editors have raised the following queries regarding figures, data, and legends. Please make sure that all requests below are completely addressed in the final version of your manuscript:
 1. Please provide the exact p values in the legends of Figures 1C, G, H, I, J, K, L, M, P, R, S, T, V, X, Z, AB; 2F, H, I; 3F, K; 4B, D, H, J; 5F, H; 6B, C, D, F, G, H, J, L; 7B, C, G, H, I, J, K, M, N; 8F, G, L.
 2. Please indicate the statistical test used for data analysis in the legends of Figures 5A, B.
 3. Please note that scale bar and its definition are missing for Figures 8K, M. This needs to be rectified.
 4. Please note that the black arrows are not defined in the legend of Figure 6A, E. This needs to be rectified.
- The manuscript section order should be corrected as follows: Title page - Abstract & Keywords - Introduction - Results - Discussion - Methods - Data Availability - Acknowledgements - Disclosure and Competing Interests Statement - References - Figure Legends - main Table(s) (if there are any) - Expanded View Figure Legends.

Please also note that as part of the EMBO publications' Transparent Editorial Process, The EMBO Journal publishes online a Peer Review File along with each accepted manuscript. This File will be published in conjunction with your paper and will include the referee reports, your point-by-point response and all pertinent correspondence relating to the manuscript. You can opt out of this by letting the editorial office know (contact@embojournal.org). If you do opt out, the Peer Review File link will point to the following statement: "No Peer Review File is available with this article, as the authors have chosen not to make the review process public in this case."

We look forward to seeing a final version of your manuscript as soon as possible. Please let us know if you have any questions and use this link to submit your revision: <https://emboj.msubmit.net/cgi-bin/main.plex>.

Kind regards,

Referee #2:

The authors have done an excellent job addressing my concerns, and the paper is significantly improved.

Minor-

The sentence in line 156 ending with reduced DAT expression is now out of order, as the reduced DAT expression doesn't appear until fig 4 now.

Referee #3:

General Summary:

In this revised version of their manuscript, Guo et al. provided additional evidence supporting the role of selenoprotein T (Selenot) on dopamine neurotransmission in hyperactive disorder (ADHD)-like behavior in mice. They offered additional behavioral experiments rescuing hyperactivity observed in Selenot f/f: Dat-cre by overexpressing Selenite via AAV transfection. This experimental procedure also rescued expression levels of NURR1 and DAT in Selenot f/f: Dat-cre mice SN and DAT expression levels in the striatum. Furthermore, Selenot knockdown seems to blunt the effect of CPA on DAT and NURR1 expression in HEK293 cells.

Opinion:

This manuscript is a complete study providing supporting evidence for the role of Selenot in dopamine function and associated behaviors.

Major concerns:

The authors addressed my major concern by providing the source of the Dat-Cre mouse line used in their experiments.

Minor concerns: The authors have addressed my previous comments and provided a well-written manuscript.

Additional comments:

- In my comment about comparing the Selenot f/f: Dat-cre vs Selenot f/f: Nestin-Cre lines to conclude that hyperactivity (and the other phenotypic findings) was solely attributable to Selenot deletion in dopamine neurons, I was taking into consideration DAT expression on no-TH positive cells reported in their original Figure 4K (now removed in this revised version). Also, Selenot is expressed on no-TH+ cells, as shown in current Figure 1B. Then, the logic provided by the authors will not apply since other cell populations, besides TH+ cells, seemed to express DAT or Selenot even in the Selenot f/f/DAT-cre mice.

-I thank the author for providing the reads per sample from their RNA seq analysis.

Please, in the following comment, note that I am not asking for new experiments but just a small explanation/discussion inserted in the manuscript:

- What was the rationale for using an AAV targeting the TH promoter region instead of targeting DAT?

- Although the knockdown of Selenot blunted the effects of CPA on SELT, DAT, and NURR in HEK293 cell cultures, these effects seem to be derived from a lower mRNA level already observed in the siSELT-2+DMSO group.

Referee #2:

The authors have done an excellent job addressing my concerns, and the paper is significantly improved.

Response: Thank you very much for your great help improving this manuscript.

Minor-

The sentence in line 156 ending with reduced DAT expression is now out of order, as the reduced DAT expression doesn't appear until fig 4 now.

Response: Thank you for the careful reading. The “with a marked reduction in DAT expression” has been removed (Line 155).

Referee #3:

General Summary:

In this revised version of their manuscript, Guo et al. provided additional evidence supporting the role of selenoprotein T (Selenot) on dopamine neurotransmission in hyperactive disorder (ADHD) -like behavior in mice. They offered additional behavioral experiments rescuing hyperactivity observed in Selenot f/f: Dat-cre by overexpressing Selenite via AAV transfection. This experimental procedure also rescued expression levels of NURR1 and DAT in Selenot f/f: Dat-cre mice SN and DAT expression levels in the striatum. Furthermore, Selenot knockdown seems to blunt the effect of CPA on DAT and NURR1 expression in HEK293 cells.

Opinion:

This manuscript is a complete study providing supporting evidence for the role of Selenot in dopamine function and associated behaviors.

Major concerns:

The authors addressed my major concern by providing the source of the Dat-Cre mouse line used in their experiments.

Minor concerns: The authors have addressed my previous comments and provided a well-written manuscript.

Additional comments:

- In my comment about comparing the Selenot f/f: Dat-cre vs Selenot f/f: Nestin-Cre lines to conclude that hyperactivity (and the other phenotypic findings) was solely attributable to Selenot deletion in dopamine neurons, I was taking into consideration DAT expression on no-TH positive cells reported in their original Figure 4K (now removed in this revised version). Also, Selenot is expressed on no-TH+ cells, as shown in current Figure 1B. Then, the logic provided by the authors will not apply since other cell populations, besides TH+ cells, seemed to express DAT or Selenot even in the Selenot f/f/DAT-cre mice.

-I thank the author for providing the reads per sample from their RNA seq analysis.

Response: Thank you very much for your great help improving this manuscript. Also thank you for your kind further explanation.

Please, in the following comment, note that I am not asking for new experiments but just a small explanation/discussion inserted in the manuscript:

- What was the rationale for using an AAV targeting the TH promoter region instead of targeting DAT?

Response: The company told us that their AAV with TH promoter has been a mature product, while the AAV with DAT promoter is less stable. We have added an explanation into the text: “Herein, the selection of TH promoter over DAT was based on the company’s instruction indicating that the AAV with TH promoter has been technically more mature.” (Lines 578-580).

- Although the knockdown of Selenot blunted the effects of CPA on SELT, DAT, and NURR in HEK293 cell cultures, these effects seem to be derived from a lower mRNA level already observed in the siSELT-2+DMSO group.

Response: Thank you for the comments. We have added a small discussion that “Notably, these effects should consider the partial contribution from the *SELENOT* knockdown per se.” (Lines 301-302).

Dear Dr. Zhu,

Congratulations on an excellent work! I am very pleased to inform you that your manuscript has now been accepted for publication in The EMBO Journal. Thank you for your comprehensive responses to the referees' comments and for addressing all editorial and formatting requests.

If you have any questions, please do not hesitate to contact the Editorial Office. Thank you for your contribution to The EMBO Journal. Working with you has been a pleasure!

Best regards,

Ioannis
